# Epithelial plasticity can generate multi-lineage phenotypes in human and murine bladder cancers

John P. Sfakianos[1,6], Jorge Daza[1,6], Yang Hu [2], Harry Anastos[1], Geoffrey Bryant[3], Rohan Bareja[2], Ketan K. Badani[1], Matthew D. Galsky[4], Olivier Elemento[2], Bishoy M. Faltas [2,5] & David J. Mulholland [3✉]

Tumor heterogeneity is common in cancer, however recent studies have applied single gene expression signatures to classify bladder cancers into distinct subtypes. Such stratification assumes that a predominant transcriptomic signature is sufficient to predict progression kinetics, patient survival and treatment response. We hypothesize that such static classification ignores intra-tumoral heterogeneity and the potential for cellular plasticity occurring during disease development. We have conducted single cell transcriptome analyses of mouse and human model systems of bladder cancer and show that tumor cells with multiple lineage subtypes not only cluster closely together at the transcriptional level but can maintain concomitant gene expression of at least one mRNA subtype. Functional studies reveal that tumor initiation and cellular plasticity can initiate from multiple lineage subtypes. Collectively, these data suggest that lineage plasticity may contribute to innate tumor heterogeneity, which in turn carry clinical implications regarding the classification and treatment of bladder cancer.

[1] Department of Urology, Icahn School of Medicine at Mount Sinai, New York, NY 10029, USA. [2] Caryl and Israel Englander Institute for Precision Medicine, Weill Cornell Medicine, New York, NY 10065, USA. [3] Division of Oncological Sciences, Icahn School of Medicine at Mount Sinai, New York 10029, USA. [4] Division of Hematology and Oncology, Icahn School of Medicine at Mount Sinai, New York 10029, USA. [5] Division of Hematology and Medical Oncology, Weill Cornell Medicine, New York, NY 10065, USA. [6]These authors contributed equally: John P. Sfakianos, Jorge Daza. ✉email: david.mulholland@mssm.edu

Recent studies have reported "intrinsic" subtypes of uro-thelial cancers based on bulk tumor gene expression pro-filing, which have been used to stratify patients according to kinetics of clinical progression, patient prognosis and response to treatment[1–3]. Although there are differences among the classification schemes developed by independent groups, at the core of each of these classifiers are tumors that express luminal-, basal-, mesenchymal-, and neuroendocrine-associated genes[2]. While the molecular classification of muscle-invasive bladder cancer represents a conceptual advance, several critical questions remain regarding these classifiers: (1) Does the lineage subtype of an individual tumor reflect the potential for tumor initiation? (2) Does the lineage subtype of an individual tumor reflect a homogeneous collection of tumor cells or is it the composite of heterogeneous populations? and (3) can the lineage subtype of a tumor change in a cell autonomous manner?

To address these questions, we applied single-cell RNA sequencing (scRNA-seq) to assess whether molecular subtyping based on bulk transcriptomes is consistent on that at the cellular level. We applied a chemically induced, transplantable mouse model of muscle invasive bladder cancer and initiated lineage transplantation strategies to assess whether heterogeneity exists and if lineage components can undergo lineage plasticity. Tumor transplant analysis demonstrated that defined populations of cells with basal, luminal, and mesenchymal qualities are capable of marked plasticity. Using the surface antigen CD49f, in conjunction with the epithelial marker Epcam, we also observed that plasticity is multidirectional with basal, luminal and mesenchymal cell populations having the ability to propagate to other tumor lineages. We further demonstrated that the heterogeneity observed in murine models of bladder cancer can be detected in human clinical samples as supported by lineage marker and mRNA expression analysis.

Collectively, our data suggest that although bladder cancers may be classified by a predominant bulk transcriptome signature, tumors can consist of multiple signatures derived from sub-populations of cells of different lineages, arising due to intrinsic heterogeneity and lineage plasticity. Our findings may help to explain the variable behavior and treatment sensitivity of bladder cancers to clinical treatments.

## Results

**Single-cell transcriptome profiling reveals multiple mRNA subtypes**. N-butyl-N-4-hydroxybutyl nitrosamine) OHBBN-induced bladder tumors are composed of heterogeneous histo-logical features including basal-like, luminal-like, and epithelial mesenchymal transition (EMT)-like cellular morphologies[4–6]. To determine whether this heterogeneity could be detected at the transcriptome level, scRNA-seq was conducted. Primary OHBBN-induced mouse bladder tumors (n = 3) were dissociated to the single-cell level, separated into viable CD45-negative and CD45-positive cells followed by 10× Chromium single-cell iso-lation and RNA-seq (Fig. 1 and Supplementary Fig. 1A). Quality-control analysis showed that nearly 96% of cells were singlets with 2.4% being recognized as doublet-high confidence and 1.9% as doublet-low confidence. Moreover, possible doublets were observed to be predominantly outside of epithelial (Epcam-positive) clusters (Supplementary Fig. 1B). Seurat was used for clustering and t-distributed stochastic neighbor embedding (tSNE) analysis, which revealed 22 cellular clusters (0–21) detected in pooled tumors and all individual tumor analysis (Fig. 1a and Supplementary Fig. 2A). Using established lineage-specific gene sets[1,2] (Table 1), we determined that CD45-negative cells comprised of 4 major epithelial clusters (clusters 3, 5, 8, and 11), 4 fibroblast (mesenchymal) clusters (0, 2, 7, and 11), 1

endothelial cluster (10), 1 lymphatic endothelial cluster (20), and 1 stromal cluster (19). In the CD45-positive population, we identified B cells (14), macrophages (1, 6, and 13), dendritic cells (4), T cells (12 and 17), monocytes (15 and 16), granulocytes (9), and mast cells (18) (Supplementary Fig. 3). Thus, scRNA-seq facilitated the identification of multiple, distinct cell populations found in CD45-negative and CD45-positive cellular fractions of OHBBN-induced mouse bladder tumors.

**Mouse bladder tumors have multi-lineage gene expression patterns**. Epithelial clusters 3, 5, 8, and 11 revealed cells with high gene expression of several lineage subtypes including basal, luminal, EMT-claudin, and neuronal differentiation but lower in P53-like and neuroendocrine marker expression (Fig. 1b). Clusters 3, 5, and 8 were noted to have high expression of basal genes (*Krt5*, *Krt6a*, *Krt14*, and *Krt16*), luminal genes (*Upk1a*, *Upk2*, *Pparg*, *Krt8*, *Krt18*, *Krt19*, *Gpx2*, *Gata3*, *Foxa1*, *Fgfr3*, and *Erbb3*), and EMT-claudin genes (*Zeb1*, *Zeb2*, *Vim*, *Snai1*, *Snai2*, *Twist1*, *Foxc2*, *Cdh1*, *Cdh2*, *Cldn1*, *Cldn3*, *Cldn4*, and *Cldn7*). Importantly, analysis of epithelial clusters revealed that constituent cells can be independently or concomitantly high in expression for basal, luminal, and EMT-claudin marker genes (Fig. 1b and Supplementary Fig. 6).

To characterize the lineage marker co-expression in the identified cell populations, we conducted several analyses. First, to visualize coinciding expression of multiple mRNA subtypes within the same tumor, we generated density plots to show cell numbers (y axis, cell density) vs. average gene expression of subtype markers (x axis, log nUMI). Using data from pooled primary mouse tumors, these plots showed that cells with the highest gene expression values (>1 nUMI) were predominantly luminal and basal with moderate gene expression of EMT-claudin, EMT-smooth muscle, and squamous subtype markers observed (0.5–1 nUMI) (Supplementary Fig. 4A). Second, to discern which cells have high expression for more than one lineage marker, we constructed a single pathway (Supplementary Fig. 5) and paired lineage tSNE plots to show the presence of bi-lineage-positive cells including basal–luminal, luminal–EMT + claudin, and basal–EMT + claudin paired subtypes (Fig. 1c). Cells with coinciding high gene expression from different subtypes are shown as red cells. Third, we constructed heatmaps of mRNA subtypes gated on individual clusters identified in single-cell sequencing analysis of CD45-negative tumor cells as either pooled (Supplementary Fig. 6A) or separated tumor data (Supplementary Fig. 6B). Focusing on epithelial clusters 3, 5, 8, and 11, we observed high gene expression from luminal, EMT-claudin, basal, and squamous subtypes. Interestingly, the concomitant high expression of genes from these three subtypes was most pronounced in clusters 3 and 8. To affirm the coinciding high expression of multiple lineage markers in cells, we constructed gene plots gated on cells with positive gene expression of *Krt5* (basal) and *Krt8* (luminal) (UMI > 0) followed by the assessment of gene expression for EMT-claudin family. We observed that *Krt5*-*Krt8*-high cells (black box) showed positive expression for *Cldn3*, *Cldn4*, and *Cldn7*, and, to a lesser extent, Vimentin (*Vim*) (Fig. 1d and Supplementary Fig. 7). Positive control (*Ck18*-*Ck8*) and negative control (*Ck8*-*Actg*) gene plots are shown (Supplementary Fig. 7).

To determine whether the above gene expression patterns existed at the protein level, we assayed for the co-distribution of basal, luminal, and EMT-claudin lineage markers in epithelial populations. As viewed in tSNE plots, *Krt5*, *Krt8*, and 3 claudin family members (*Cldn3*, *Cldn4*, and *Cldn7*) localized in part to epithelial clusters (3, 5, and 8) with *Krt8* and *Cldn3* also showing high expression in clusters 2 and 11 (Fig. 2a). Using triple labeling

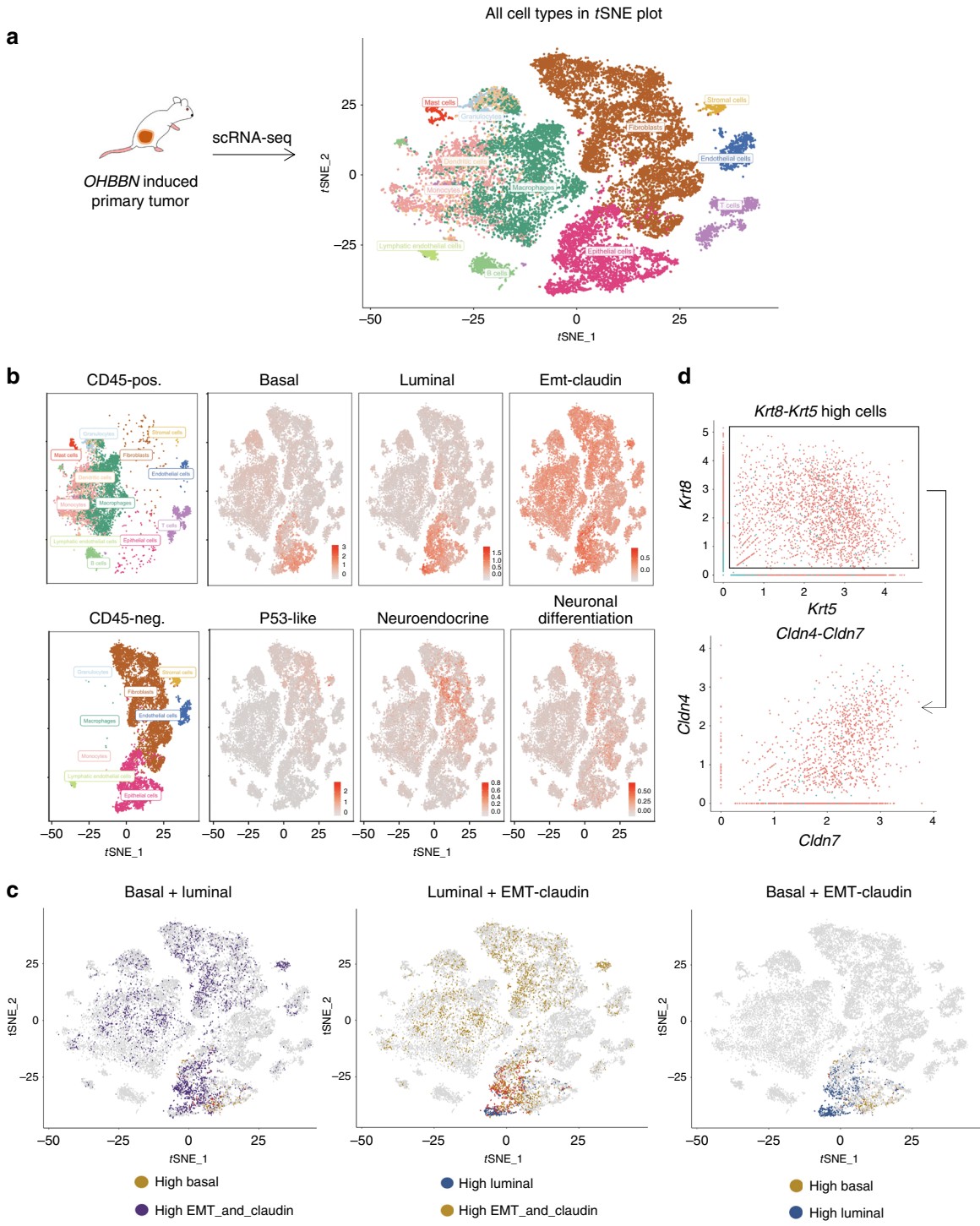

**Fig. 1 Single-cell transcriptome profiling of OHBBN-induced mouse bladder tumors. a** tSNE plots showing a total of 22 clusters identified from CD45-negative and CD45-positive fractions isolated by flow cytometry from OHBBN-induced primary bladder tumors (pooled tumors, $n = 3$ tumors). **b** tSNE plots showing the proximity of cells with high expression of genes belonging to basal, luminal, and EMT-claudin lineage subtypes within Epcam-positive defined populations. **c** Single-cell gene plots gated on cells with high *Krt5* and *Krt8* gene expression followed by the assessment of EMT-claudin genes (*Cldn4*, *Cldn7*). Cells with *Krt5* > 0 and *Krt8* > 0 expression were gated as *Krt5-Krt8* high, whereas cells with *Krt5* = 0 and *Krt8* = 0 were gated as *Krt5-Krt8* low. Axis units are log (UMI) or transformed transcripts per cell. **d** tSNE plots showing the presence of bi-lineage-positive cells using gene expression overlays from (left) basal + luminal, (middle) luminal + EMT-claudin, and (right) basal + EMT-claudin subpopulations. Tumor identification and cell numbers sequenced were as follows: tumor 4950 (CD45-neg = 2939 cells, CD45-pos = 1307 cells), tumor 8524 (CD45-neg = 6119 cells, CD45-pos = 2736 cells), and tumor 8525 (CD45-neg = 5068 cells, CD45-pos = 7564 cells). Genes assessed in tSNE plots are shown in Table 1. Top up and down genes are shown in Supplementary Data 1.

**Table 1 Examples of lineage genes used for analysis.**

| Luminal | CYP2J2 | ERBB2 | ERBB3 | FGFR3 | FOXA1 | GATA3 | GPX2 | KRT18 | KRT19 |
|---|---|---|---|---|---|---|---|---|---|
| | KRT20 | KRT7 | KRT8 | PPARG | XBP1 | UPK1A | UPK2 | | |
| EMT and smooth muscle | PGM5 | DES | C7 | SRFP4 | COMP | SGCD | | | |
| EMT and claudin | ZEB1 | ZEB2 | VIM | SNAI1 | TWIST1 | FOXC2 | CDH2 | CLDN3 | CLDN7 |
| | CLDN4 | CDH1 | SNAI2 | | | | | | |
| Basal | CD44 | CDH3 | KRT1 | KRT14 | KRT16 | KRT5 | KRT6A | KRT6B | KRT6C |
| P53-like | ACTG2 | CNN1 | MYH11 | MFAP4 | PGM5 | FLNC | ACTC1 | DES | PCP4 |
| Squamous | DSC1 | DSC2 | DSC3 | DSG1 | DSG2 | DSG3 | S100A7 | S100A8 | |
| Immune | CD274 | PDCD1LG2 | IDO1 | CXCL11 | L1CAM | SAA1 | | | |
| Neuroendocrine | CHGA | CHGB | SCG2 | ENO2 | SYP | NCAM1 | | | |
| Neuronal differentiation | MSI1 | PLEKHG4B | GNG4 | PEG10 | RND2 | APLP1 | SOX2 | TUBB2B | |
| Downregulated CIS | CRTAC1 | CTSE | PADI3 | | | | | | |
| Upregulated CIS | MSN | NR3C1 | | | | | | | |
| Cancer stem cell | CD44 | KRT5 | RPSA | ALDH1A1 | | | | | |

immunofluorescence, expression of Ck5, Ck8, and Cldn7 was assessed at low and high magnifications, allowing for the detection of single (arrowheads)-, double (open arrows)-, and triple-lineage marker-positive cells (dashed, open arrows) (Fig. 2b, c and Supplementary Fig. 8). In regions of carcinoma in situ or lumen adjacent regions, we observed a preponderance of double positive cells (Fig. 2, rows 1–2). Conversely, in areas of poorly differentiated cancer, cells positive for basal, luminal, and EMT-claudin markers were more prevalent (row 3). Interestingly, we observed cancer regions that were claudin-high, -mid, and claudin-low in expression (rows 1–3). Such expression patterns were consistent between the three claudin markers tested including Cldn3, Cldn4, and Cldn7. Collectively, these data reveal that in OHBBN-induced mouse primary bladder tumors, multiple lineage subtypes can be detected at the transcriptomic and protein levels.

**Human bladder tumors have multi-lineage gene expression patterns.** To determine whether multiple lineage markers are expressed within individual human tumors, scRNA-seq was conducted on sorted CD45-negative cell populations. Using two muscle-invasive human bladder tumors (tumors 357 and 359), unsupervised clustering analysis identified eight clusters (Fig. 3a and Supplementary Fig. 2C). Density distribution analysis of normalized lineage gene expression showed these tumors to have diverse average gene expression with tumor 357 having high luminal subtyping and 359 showing less luminal gene expression (tumor 357 = 3306 cells sequenced, tumor 359 = 540 cells sequenced). In both cases, there was positive lineage scores (average gene expression, log nUMI) for luminal (blue), basal (orange), and EMT-claudin (light green) marker expression (Fig. 3b and Supplementary Fig. 4B). Conversely, in both tumors, gene expression of smooth muscle markers (dark green) showed a high density of cells with low or negative log nUMI values that did not overlap with the luminal, basal, and EMT-claudin cell populations (Supplementary Fig. 4B). Tumors 357 and 359 were both low in expression for P53-like subtype genes or those associated with the neuroendocrine lineage (Supplementary Fig. 9). Both 357 and 359 displayed a mixed composition of basal, luminal, and EMT-claudin-associated genes including individual cells with high expression of luminal and EMT-claudin genes (Fig. 3c). Similar to gene plot analysis in mouse tumors, cells double positive for CK8 (luminal) and CK14 (basal) (CK8-CK14 high) expression also showed positive expression of CLDN3, CLDN4, and CLDN7. Positive control (CK8-CK18) and negative control (KRT5-ACTG) gene plots are shown (Fig. 3d and Supplementary Fig. 7B, C). At the histological level, HE analysis for tumors 357 and 359 showed diverse pathology including

epithelial and mesenchymal-like morphologies (Supplementary Fig. 9I).

Human primary bladder tumors were assessed for positive immunostaining of basal (Ck5, p63), luminal (Ck8), and EMT-claudin (Cldns 4, 5, or 7) markers from which we identified tumor regions being basal only, luminal only, and mixed basal–luminal in composition. (Supplementary Fig. 10). Lineage heterogeneity was also detected in bladder cancers that had metastasized to the lymph node (LN) ($n = 17$) where we identified single lineage-positive basal (4/17), luminal (6/17), and mixed basal–luminal lineage-positive LNs (7/17) (Supplementary Fig. 10B, C). Mixed metastatic lesions comprised single cells (left), acinar-like expansions (middle), and regionally defined independent lesions (right). In each type of mixed-lineage LN metastasis, cells single (CK5$^+$ or CK8$^+$) and double positive (CK5$^+$CK8$^+$) were observed. Mixed basal–luminal LN metastasis could be triple-lineage positive as exemplified by triple stains for Ck5-Ck8-Cldn7 or p63-Ck5-Cldn7 (Supplementary Fig. 10D, E). Together, these single-cell analyses show that human muscle-invasive bladder tumors contain individual epithelia with gene expression patterns characteristic of different mRNA subtypes, and that these cells can be concomitantly or independently high in basal-luminal-EMT-claudin marker expression.

**Bulk transcriptome profiling of lineage subtypes in mouse bladder tumors.** To corroborate our findings and to assess changes in lineage populations with respect to normal urothelium, we used a publicly available data set for primary tumors in OHBBN-treated mice (Illumina mouse-6 v1.0 expression beadchip, GEO accession: GSE21636)[7]. We compared normal mouse urothelia ($n = 5$) with OHBBN-induced bladder tumors ($n = 5$) and searched for differentially expressed genes. Using genes found in lineage and subtyping related populations[2], 54 genes with expression values in the mouse data set were identified. We derived mouse-specific signatures to facilitate clustering of tumors based on basal, luminal, neuronal, extracellular matrix and smooth muscle, mesenchymal, EMT-claudin, and squamous and immune markers showing clear separation between normal bladder and those with cancer (Fig. 4a and Supplementary Fig. 11). Genes overexpressed in OHBBN-treated samples, relative to the normal urothelial, belonged to different mRNA subtypes including basal (Cd44, Cdh3, Col17a1, Egfr, Hif1a, Itga6, Myc, Nfkb, and Stat3), luminal (Cd24a, Cyp2j6, Fgfr3, Gpx2, Trim24, and Xbp1), neuronal differentiation (Chga and Sox2), EMT-claudin (Vim, Twist, Snai1, Snai2, Smn1, Cldn3, Cldn4, Cldn7, and Smn1), squamous (Dsc3 and Trp63), and immune (Cxcl11, Pdcd1lg2, and Saa3). As identified with scRNA-seq analysis of mouse and human tumors, bulk RNA-seq showed

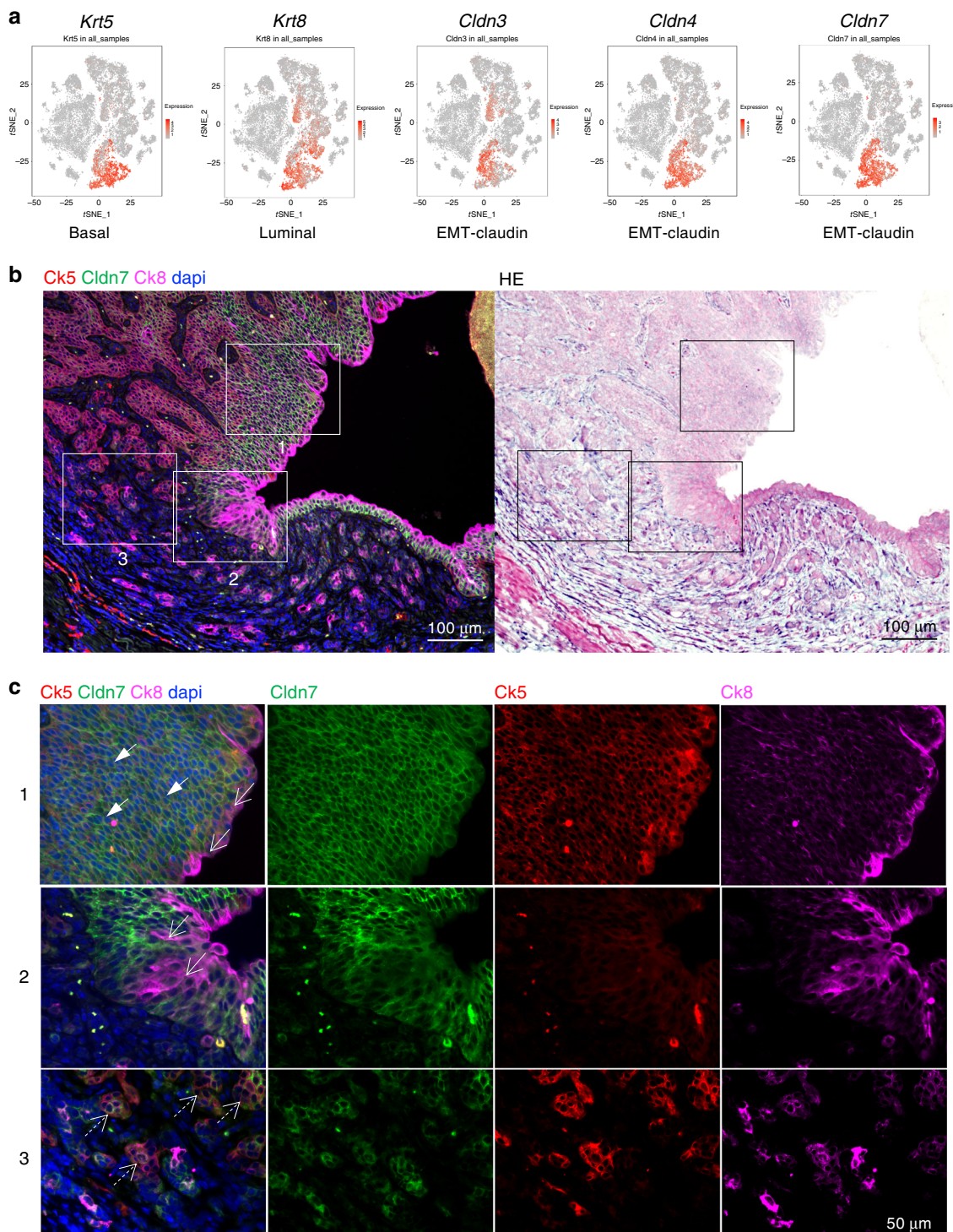

**Fig. 2 Detection of OHBBN-induced bladder cancer cells with single-, double-, and triple-lineage marker-positive cells. a** Single-cell RNA-seq analysis showing the presence of epithelial cells high in basal (*Krt5*), luminal (*Krt8*), and EMT-claudin (*Cldn3*, *Cldn4*, and *Cldn7*) marker gene expression. **b** Low-magnification triple immunofluorescence detection of Ck5 (*Krt5*), Ck8 (*Krt8*), and Cldn7 (*Cldn7*), and the corresponding hemotoxylin & eosin (re-stain of IF slide). **c** High-magnification views of insets from **b** showing instances of tumor cells co-expressing Ck5-Cldn7-Ck8$^{low}$ (solid arrows), Ck8$^{hi}$-Cldn7 (open arrows), and Ck5-Cldn7-Ck8 (open dashed arrows). Each experiment was performed at least in triplicate.

OHBBN tumors to cluster into several subtypes. To compare lineage pathways in the tumor samples analyzed, we calculated and plotted *z*-scores to identify outlier genes for each lineage (color and boxes) including basal (*Itga6*, *Cd44*, and Col17a1), squamous (*Tgm1*), luminal (*Gpx2* and *Erbb2*), EMT-claudin (*Snail1*, *Snail2*, *Cldn7*), Neuronal differentiation (*Sox2* and *Chga*),

Immune (Saa3), and luminal (*Gpx2* and *Upk2*) shown together (Supplementary Fig. 11B) or as independent lineages (Supplementary Fig. 11C). Thus, using a second expression analysis approach, we have demonstrated that genes belonging to multiple clinical mRNA subtypes can be concomitantly overexpressed in OHBBN-induced bladder tumors.

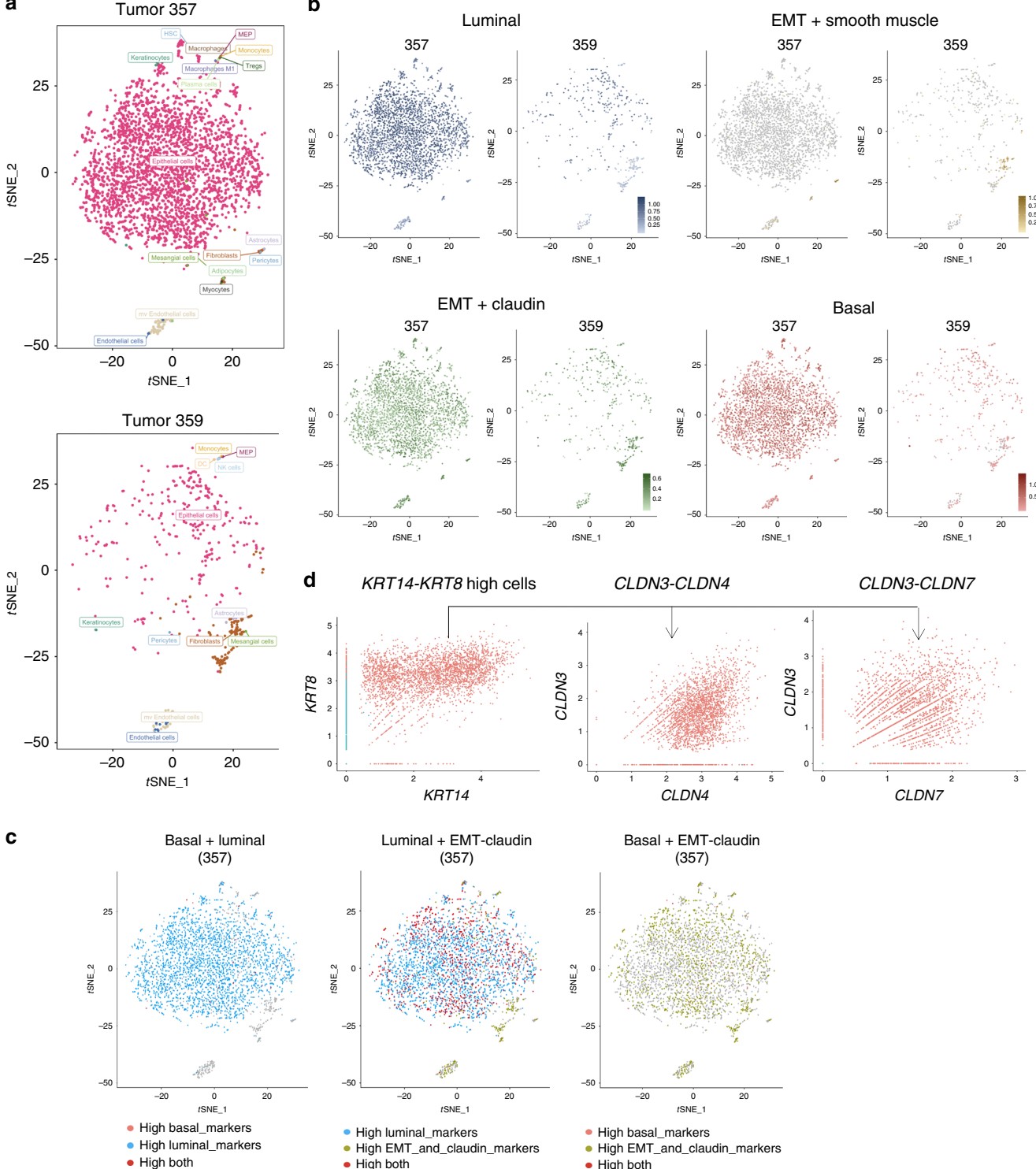

**Fig. 3 Single-cell transcriptome profiling of human MIBCs. a** tSNE plots showing clustering analysis from two independent muscle-invasive bladder tumors (357 and 359) identified using sorted CD45-negative cells. **b** tSNE plots from two bladder tumors showing the coinciding expression of cells with high gene expression from basal, luminal, and EMT-claudin subtypes. **c** tSNE plots in tumor 357 showing the presence of luminal + EMT-claudin, bi-lineage-positive cells using gene expression overlays. **d** Single-cell gene plots using *KRT14-KRT8*-high gene expressing cells followed by assessment for positive expression of EMT-claudin genes (*CLDN3*, *CLDN4*, and *CLDN7*). Cells with *KRT14* > 0 and *KRT8* > 0 expression were gated as *KRT14-KRT8* high. Cells with *KRT14* = 0 and *KRT8* = 0 were gated as *KRT14-KRT8* low. Axis units are log (UMI) or transformed transcripts per cell. Genes used in tSNE plots are shown in Table 1. Top up and down genes for human tumors are in Supplementary Data 2 and 3.

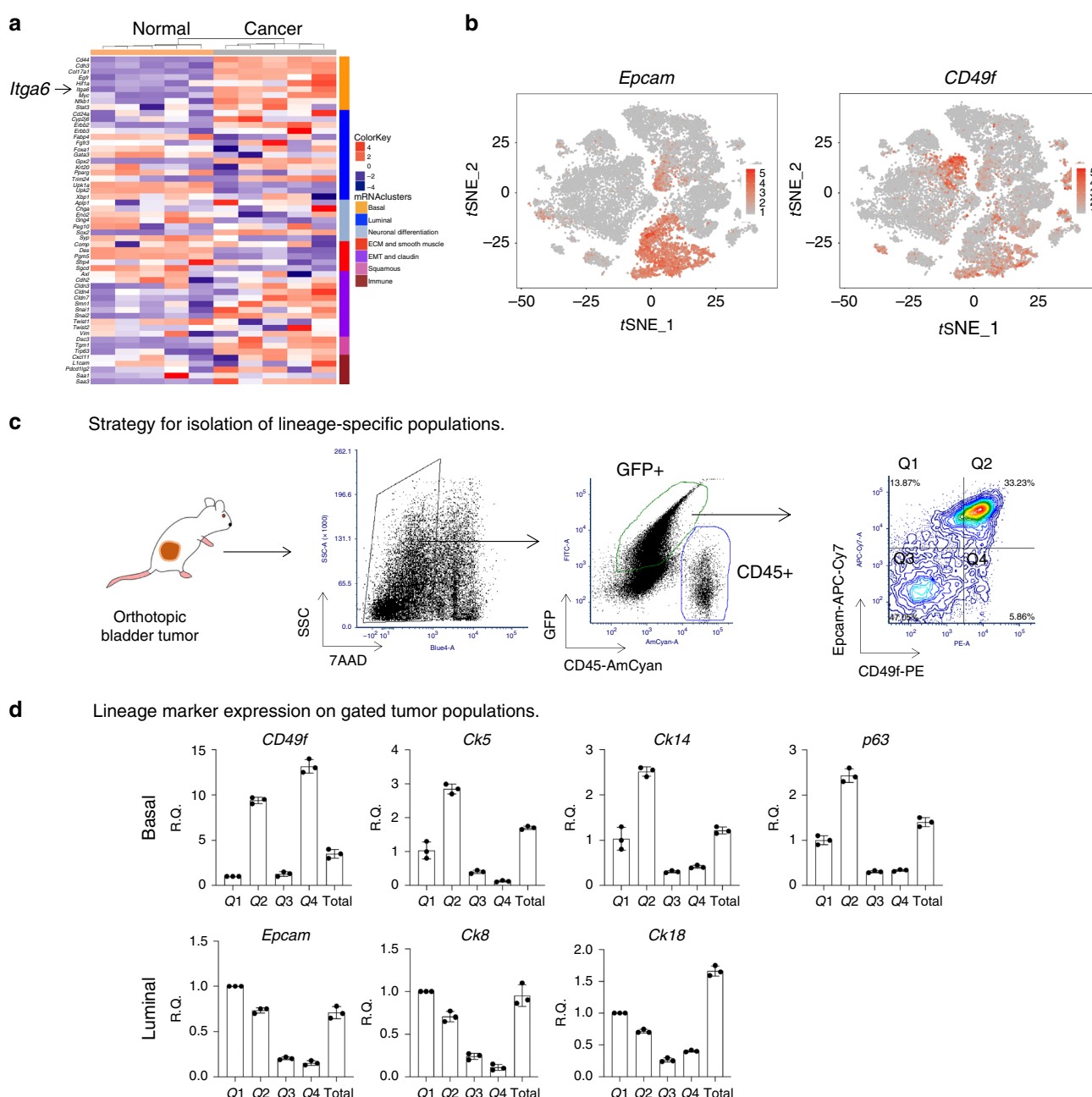

**Fig. 4 Identification of CD49f (*Itga6*) as a differentially expressed cell surface marker in OHBBN-induced bladder tumors. a** Bulk RNA-seq analysis showing differentially expressed genes belonging to various lineage subtypes in the normal bladder (*n* = 5) and OHBNN induced bladder cancer (*n* = 5). The surface antigen, CD49f (*Itga6*, arrows), is overexpressed in bladder tumors and found within the basal mRNA cluster. **b** tSNE cell clusters 3, 5, 8, and 11 defining Epcam (left) and CD49f (*Itga6*) expression. **c** Strategy for discriminating GFP-positive, CD45-negative OHBBN-induced cancer cells from non-tumor host cells in FVB/NJ-transplanted mice. Tumor cells are evaluated for expression of Epcam and CD49f, and define lineage populations within defined quadrants (Q1, Q2, Q3, and Q4). **d** Differential gene expression for lineage subtypes using Epcam-CD49f expression in cell populations of GFP-positive; CD45-negative tumor cells sorted from quadrants Q1, Q2, Q3, and Q4. Individual data points are represented by circles, the error bars represent mean ± SD of the three independent replicates. Each experiment was performed at least in triplicate.

**CD49f defines bladder tumor cells with multiple lineage signatures.** scRNA-seq analysis identified populations of mouse and human epithelia with high expression of basal, luminal, and EMT-claudin-associated genes. Thus, we hypothesized that such cells may have the potential to undergo lineage plasticity or conversion between subtypes. To model this, we isolated and surgically transplanted specific lineages found in OHBBN tumor cells using the surface antigens CD49f and Epcam. Using bulk RNA-seq analysis, we showed CD49f (*Itga6*) to be expressed at markedly higher levels in tumors vs. normal mouse bladders (Fig. 4a, left arrow). scRNA-seq showed cells with high expression of *Itga6* to be located in epithelial cell clusters 3, 5, 8, and 11 within the Epcam-positive population (Fig. 4b). To validate the protein expression of CD49f, we surveyed OHBBN primary mouse tumors ($n = 10$), OHBBN mouse derived allografts ($n = 17$) (Fig. 4c and Supplementary Fig. 12), and human bladder tumors ($n = 17$) (Supplementary Fig. 12). Gating on CD45-negative cells, we discriminated two predominant tumor cell populations including Epcam$^{high}$CD49f$^{high}$ (quadrant 2, Q2) and Epcam$^{low}$CD49f$^{low}$ (Q3), and minor populations in Q1 and Q4. In tumor mouse allografts, we also observed a high content of Epcam$^{high}$CD49f$^{high}$ and Epcam$^{low}$CD49f$^{low}$ cells (Supplementary Fig. 12A, B) validated by representative fluorescence-activated cell sorting (FACS) staining controls (Supplementary Fig. 12C). Analysis of human primary bladder tumors ($n = 17$, high grade, muscle invasive) revealed more heterogeneity including those with little or no CD49f expression (<2%, $n = 3$), low CD49f expression (2-10%, $n = 6$), and those with moderate to high expression (>10%, $n = 7$). For CD49f-positive tumors, we observed 7/17 that showed Epcam and CD49f co-expression (Supplementary Fig. 12D). With this, we considered whether the CD49f-Epcam marker combination could discriminate basal, luminal, and mesenchymal cell populations in primary or

transplanted OHBBN-induced bladder tumors. By quantitative PCR (q-PCR), we detected that Epcam$^{high}$CD49f$^{high}$ (Q2) cells had elevated gene expression of multiple basal markers (*Ck5*, *Ck14*, and *p63*) (Fig. 4d). Luminal marker expression was highest in Epcam$^{high}$CD49f$^{low}$ (Q1) cells, whereas the highest expression of mesenchymal lineage markers (*Sma*, *Vim*, and *Axl*) was observed in Epcam$^{low}$CD49f$^{low/high}$-expressing cells (Q3). These analysis suggest that such non-tumorigenic cells are diluted out during tumor growth and transplantation from primary to green fluorescent protein (GFP)-negative hosts. Our findings also show that CD49f is overexpressed in OHBBN bladder tumors and can be used in conjunction with Epcam to discriminate and isolate cells with basal, luminal, and EMT-like qualities (Table 2).

**Modeling lineage plasticity using in-vivo transplantation of bladder cancer subpopulations.** We identified multiple lineage subtypes within individual mouse bladder tumors. As such, we considered whether these cell populations remained static or had the potential for progression-induced lineage plasticity. For this, we applied an in-vivo model amenable to the surgical implantation of isolated cells and subsequent re-analysis of tumor outgrowths after progression. For this, *Tg(CAG-luc-eGFP)* FVB/NJ mice were treated with OHBBN to induce high-grade and invasive primary tumors shown at the gross and histological levels (Supplementary Fig. 13A). Despite this, invasive primary tumors were slow to develop (16–20 weeks), had asynchronous growth kinetics, and failed to show metastasis to organs including the liver. Using surgical implantations of isolated primary cells, we generated tumor outgrowths with reproducible and synchronous kinetics in the bladder and liver. As tumor cells were enhanced GFP (eGFP) and luciferase tagged, we could distinguish transplanted cells from non-tumor host populations (Figs. 4c and 5b).

**Table 2 Primer sequences for q-PCR analysis.**

| Markers | Name | Gene | Forward | Reverse |
|---|---|---|---|---|
| EMT | p63 | *Trp63 or Tp63* | 5′-GAAGGCAGATGAAGACAGCA-3′ | 5′-GGAAGTCATCTGGATTCCGT |
| | ΔN-p63 | *DNp63* | 5′-TCTGATGGCATTTGACCCTA-3′ | 5′-TACCAACAGATGGGAAGCAA |
| | CK5 | *Krt5* | 5′-ACCTTCGAAACACCAAGCAC-3′ | 5′-TTGGCACACTGCTTCTTGAC |
| | CK14 | *Krt14* | 5′-GACTTCCGGACCAAGTTTGA-3′ | 5′-CCTTGAGGCTCTCAATCTGC |
| | CD44 | *Cd44* | 5′-AGCAGCGGGCTCCACCATCGAG A-3′ | 5′-TCG GAT CCA TGA GTC ACA GTG |
| | CD49f | *Itga6* | 5′-AGAGACATGAAGTCCGCGCA-3′ | 5′-ACCTTCCCCAGATCATCATAG |
| | CK8 | *Krt8* | 5′-ATCGAGATCACCACCTACCG-3′ | 5′-TGAAGCCAGGGCTAGTGAGT |
| | CK18 | *Crt8* | 5′-CTGGAGGATGGAGAAGATTT-3′ | 5′-CTTTTATTGGTCCCTCAGTT |
| | Gata3 | *Gata3* | 5′-CTCGGCCATTCGTACATGGAA-3′ | 5′-GGATACCTCTGCACCGTAGC |
| Basal | E-Cad | *Cdh1* | 5′-CAGGTCTCCTCATGGCTTTGC-3′ | 5′-CTTCCGAAAAGAAGGCTGTCC |
| | Epcam | *Epcam* | 5′-TTGCTCCAAACTGGCGTCTA-3′ | 5′-ACGTGATCTCCGTGTCCTTGT |
| | Vimentin | *Vim* | 5′-CGGCTGCGAGAGAAATTGC-3′ | 5′-CCACTTTCCGTTCAAGGTCAAG |
| | SMA | *Acta2* | 5′-CTGACAGAGGCAACCACTGAA-3′ | 5′-CATCTCCAGAGTCCAGCACA |
| | Fibronectin 1 | *Fn1* | 5′-AGCAGTGGGAACGGACCTAC-3′ | 5′-ACGTAGGACGTCCCAGCAGC |
| | GAPDH | *Gapdh* | 5′-TGAACGGGAAGCTCACTG G-3′ | 5′-TCCACCACCCTGTTGCTGTA-3′ |
| | Snail1 | *Snai1* | 5′-AAGATGCACATCCGAAGC-3′ | 5′-ATCTCTTCACATCCGAGTGG-3′ |
| | Snail2 | *Snai2* | 5′-GATGTGCCCTCAGGTTTGAT-3′ | 5′-GGCTGCTTCAAGGACACAT-3′ |
| | tdTomato | *tdTomato* | 5′-ACCATCGTGGAACAGTACGAG-3′ | 5′-CTTGAAGCGCATGAACTCTTT-3′ |
| | EGFP | *eGFP* | 5′-GAAGCAGCACGACTTCTTCAA-3′ | 5′-AAGTCGATGCCCTTCAGCTC-3′ |
| Luminal | AXL | *Axl* | 5′-GGAGGAGCCTGAGGACAAAGC-3′ | 5′-ACAGCATCTTGAAGCCAGAGTAGG-3′ |
| | Axl | *Axl* | 5′-AGGCTCATTGGCGTCTGTT-3′ | 5′-ATCGCTCTTGCTGGTGTAG-3′ |
| | GAS6 | *Gas6* | 5′-CGGCATTCCCTTCAAGGAGAGT-3′ | 5′-CTCAACTGCCAGGACCACCAACT-3′ |
| | PD-1 | *Pdcd1* | 5′-GCAATCAGGGTGGCTTCT-3′ | 5′-TTGGCTCAAACCATTACAGA-3′ |
| | PD-L1 | *Cd274* | 5′-GATTCAGTTTGTGGCAGGAGA-3′ | 5′-GTATGGGGCATTGACTTTCA-3′ |
| | TIM-3 | *Havcr2* | 5′-CCACGGAGAGAAATGGTT C-3′ | 5′-CATCAGCCCATGTGGAAAT-3′ |
| | CD45 | *Ptprc* | 5′-CCTCCAACTCCTGTATGAAGGA-3′ | 5′-GAGGGCAAAGTGGATGTCTATG-3′ |
| | cd45 | *Ptprc* | 5′-CTTGACTTGTCCATTCTGGG-3′ | 5′-GGACAACGCAGACTCTCACATT-3′ |
| | IL-2 | *Il2* | 5′-CCTGAGCAGGATGGAGAATTACA-3′ | 5′-TCCAGAACATGCCGCAGAG-3′- |
| | IL-6 | *Il6* | 5′-CTCTGGGAAATCGTGGAAAT-3′ | 5′-CCAGTTTGGTAGCATCCATC-3′ |

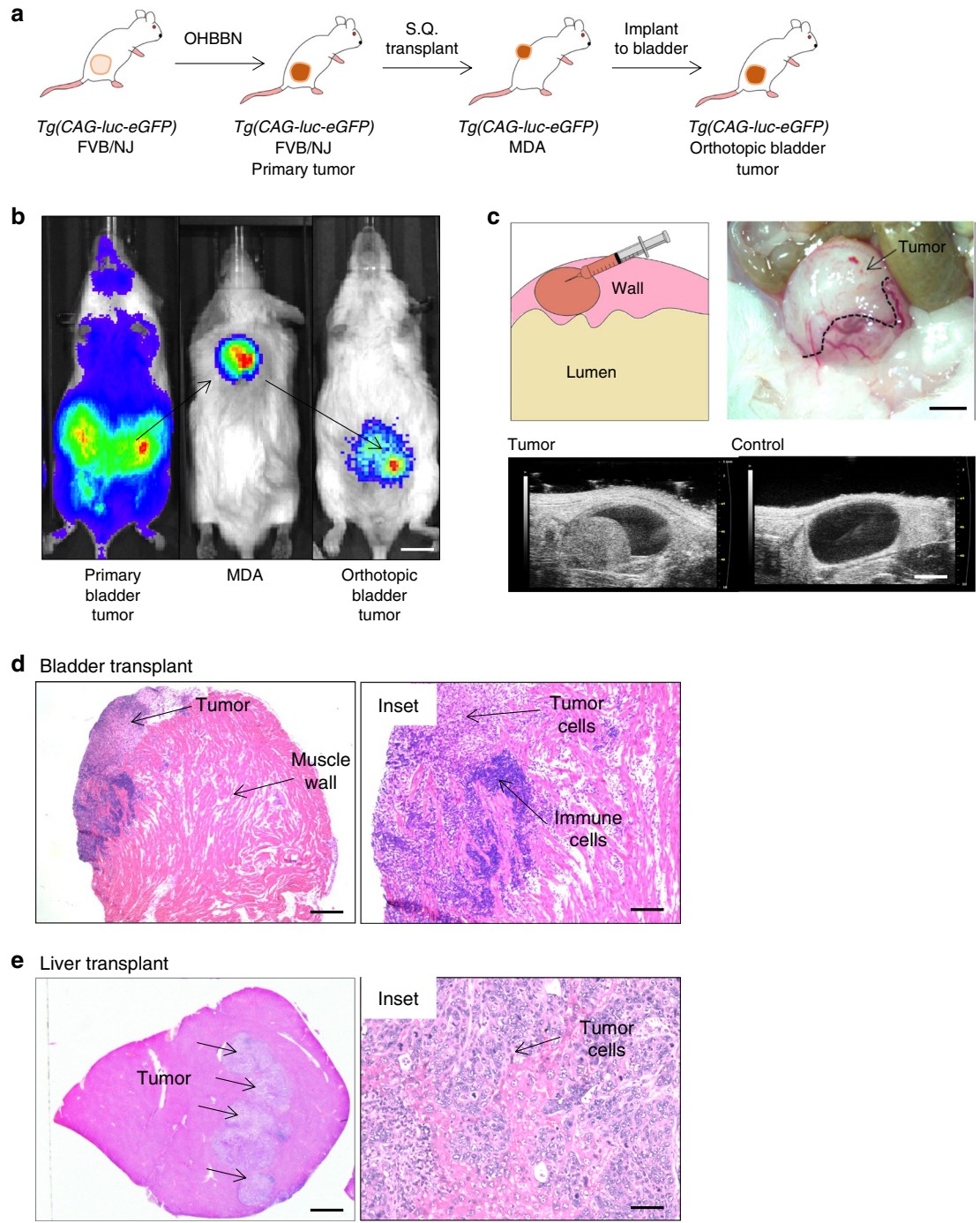

**Fig. 5 Modeling lineage plasticity using in-vivo transplantation of mouse bladder cancer subpopulations. a** Schematic showing the propagation of carcinogen induced, transplantable bladder cancers in FVB/NJ mice. FVB/NJ *Tg(CAG-luc-eGFP)* mice were treated with OHBBN followed by surgical resection of primary tumors and their subcutaneous (S.Q.) transplantation to FVB/NJ mice. Established primary tumors and mouse derived allografts (MDAs) were successfully passaged, used for surgical implantations to the bladder wall or liver. **b** Bioluminescence imaging of a primary tumor-bearing mouse (left), derived S.Q. tumors (middle), and mouse with implanted tumor cells to the bladder muscle wall (right) (Bar = 7 mm). **c** Schematic, gross image, and ultrasound images of tumor cell implants to the bladder muscle wall (top bar = 4 mm, bottom bar = 3 mm). **d** Histology of tumors transplanted to the bladder showing immune cells and invasive pathology (left bar = 500 μM, right bar = 150 μM). **e** Histology of tumor cells transplanted to the liver showing lesion formation and high-grade pathology (left bar = 2 mm, right bar = 50 μM). These studies were reproduced with at least 50 biological replicates.

Successful implants were achieved with as a few as 2–5K cells and typically formed macroscopic lesions 3–6 weeks post injection (Fig. 5c, upper panels) and similar to primary *Tg(CAG-luc-eGFP)* FVB/NJ tumors were readily visible by ultrasound imaging

(Fig. 5c). Histological analysis of transplants revealed high-grade and invasive pathology coinciding with a high content of immune infiltration (Fig. 5d, e). To affirm that basal, luminal, and mesenchymal populations were expressed in transplant tumor

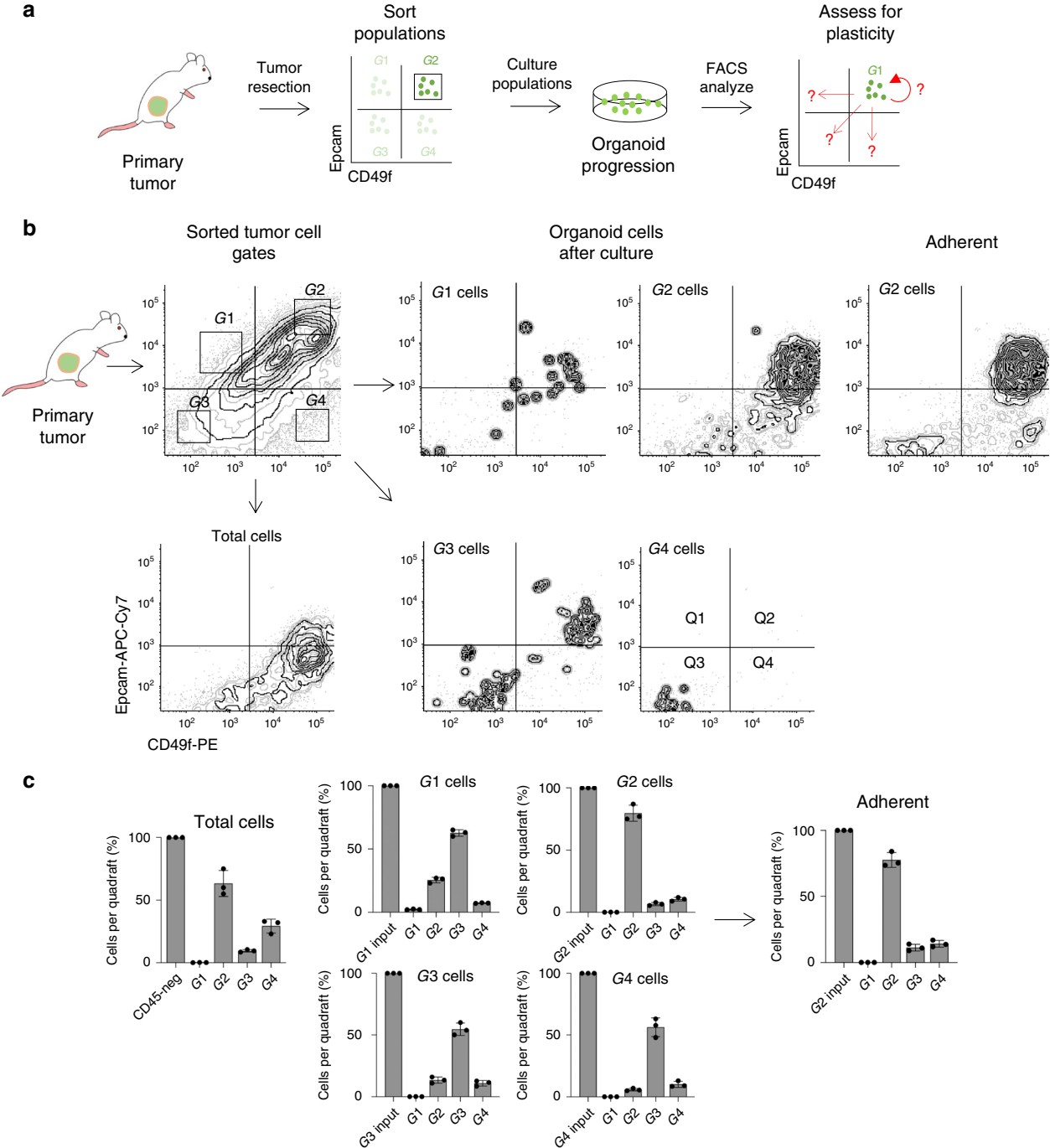

**Fig. 6 Lineage plasticity of mouse bladder cancer cell populations in vitro. a** Strategy to evaluate the plasticity in Epcam and CD49f expression using sorted populations of OHBBN-induced primary bladder cancer cells cultured as organoids. **b** Flow cytometry plots showing isolated cell populations of tumor cells (left), grown in vitro for 10–15 days, and analyzed for changes in Epcam and CD49f expression. The effects of switching to adherent conditions are shown for gate two (G2) cultured cells (top right) and total CD45-negative cells grown continuously in suspension (bottom left). **c** Graphs showing redistribution of individual sorted cell populations before (input profile) and after in-vitro culture. Individual data points are represented by circles, the error bars represent mean ± SD of the three independent replicates. Each experiment was performed at least eight times.

cells, we conducted lineage co-immunofluorescence immuno-typing using transgenic eGFP. Within single transplants, we observed lineage markers co-expressing basal (p63+ GFP+, CK5+ GFP+), luminal (CK8+ GFP+), and mesenchymal (Sma+ GFP+, Vim+ GFP+) lineages at low and high magnification (Supplementary Fig. 13C). These data support the development of a mouse model amenable for the transplantation and tracking of bladder cancer cells using immune competent hosts.

**CD49f defines bladder tumor formation and lineage differentiation.** We identified CD49f to be highly expressed in epithelial clusters of mouse bladder tumors. Thus, we tested whether its expression could define populations of cancer cells capable of tumor formation and lineage plasticity. Using flow cytometry, we isolated cell populations based on expression of Epcam and CD49f from primary or P0 tumors gating on CD45-negative; GFP-positive cells (Figs. 4c and 6a). In general, each sorted

population was enriched twice by flow cytometry with isolated cell populations being confirmed using five independent quality checks to have purities of 96–99% when reference to fluorescence minus one (FMO) staining controls (Supplementary Fig. 14A). Sorted cells were cultured at varying densities (5, 10, or 50 K cells per well of a 12-well plate) ($n = 3$ tumors) in defined organoid media conditions[8] and allowed to progress in suspension (non-adherent) culture conditions. After 10–15 days, cultures were assessed for changes in Epcam and CD49f expression. Although cultured total GFP$^{pos}$CD45$^{neg}$ cell populations maintained similar Epcam-CD49f expression to in-vivo primary tumor cells, isolated populations showed pronounced expression changes in surface antigens relative to the original, input cell populations. This included gate 2 expressing cells shifting to quadrants 3 and 4 (G2 → Q3, Q4), as well as (G1 → Q2, Q3, Q4), (G3 → Q2, Q3), and (G4 → Q2, Q3) (where G = input gate and Q = quadrant). The most pronounced shifts were Epcam$^{hi}$CD49f$^{low}$ (G1)-expressing cells shifting to Q3, Epcam$^{low}$CD49f$^{low}$ (G3) cells moving to Q2, and Epcam$^{low}$CD49f$^{high}$ (G4) moving to Q3 (Fig. 6b, c). Changes in Epcam-CD49f expression were also observed when transferring cells from suspension conditions to adherent growth conditions on Matrigel. These data show that OHBBN bladder cancer cells can undergo plasticity in surface antigen and lineage marker expression in a manner that is cell autonomous and independent from the stromal and immune tumor microenvironments.

**CK5 expression defines a CD49f-positive tumor-initiating cell population.** The above in-vitro observations prompted us to consider whether CD49f could define the in-vivo tumor-initiating capacity of OHBBN-induced bladder cancer cells. To do this, we used a *Krt5*-tomato reporter mouse line to positively identify and trace the lineage fate of CK5$^{high}$ (basal cells) in normal and tumor-bearing bladders. When *Krt5-Cre$^{ERT2}$;Tdt$^{LSL}$* mice were pulsed with tamoxifen (TAM; 4× doses, intraperitoneally, 250 mg/kg), the *Krt5* promoter was activated to induce tomato expression and subsequent lineage tracing ($n = 4$) along with non-TAM-treated controls ($n = 4$). Gating on CD45$^{neg}$;Tdt$^{high}$ cells, we observed 82.7% to be Epcam$^{high}$;CD49f$^{high}$ and 15% to be Epcam$^{high}$;CD49f$^{mid}$ in expression (Fig. 7a), a finding consistent with our q-PCR data showing high Ck5 gene expression in CD49f$^{high}$ cells (Fig. 4d). When TAM-treated *Krt5-Cre$^{ERT2}$; Tdt$^{LSL}$* mice ($n = 4$) were induced to form cancer with OHBBN and subsequently assessed for Epcam-CD49f expression we observed significant distribution of tomato positive cells to all gates (G1 = 31.4%, G2 = 39.5%, G3 = 20.5%, G4 = 6.9%, $n = 4$) (Fig. 7a, bottom). These data affirm that CK5-positive cells are predominantly high in CD49f expression in the normal bladder but when induced to form cancer are capable of differentiating to populations with low and high expression of CD49f and Epcam.

**Basal and EMT-like cells are efficient initiators of tumorigenesis.** We showed that cells with basal, luminal, and mesenchymal lineage marker expression can co-exist in single bladder tumors. To determine the relative efficiency for each subpopulation to initiate tumorigenesis, we used in-vitro and in-vivo strategies. For in-vitro measurements, GFP$^{pos}$CD45$^{neg}$ cells were isolated from OHBBN primary or P0 tumors according to CD49f-Epcam expression using defined gates, plated in droplets of Cultrex, and assessed for organoid formation at different plating densities (1000, 2000, 5000, or 50,000 cells, where counted organoids > 100 μm diameter) (Supplementary Fig. 14A). We observed significant differences in organoid formation between subpopulations detectable at 5000 and 50,000 plating densities where G3 > G2 > G4 > G1 ($p < 0.05$, Student's *t*-test). Significant differences were

not observed using 1000 or 2000 cell plating conditions ($p > 0.05$, Student's *t*-test) with the exception of Gate 3 (2000) (*). For in-vivo tumor analysis, subpopulations were isolated from primary or P0 tumors and implanted cells to the bladder muscle wall of FVB/NJ mice. With this strategy, we detected tumor growth (defined by a visible mass, ~0.05 cm³) for all lineages but with marked differences in the efficiency in tumor initiation between lineages (G3 > G2 > G4 ≅ G1) most notable using 10,000 and 50,000 cell implants (Supplementary Fig. 14C). Cohorts of implants using 2000 cells, in general, failed to form palpable masses (NG = no growth) even after 2–3 months but in certain instances demonstrated the retention of viable cells within the Cultrex deposit (NG*). Flow cytometry analysis of in-vivo transplants after progression recapitulated in-vitro trends, where G2 transplants resulted in cells redistributing to all quadrants (Q1 = 10.0%, Q2 = 37.0%, Q3 = 37.0%, and Q4 = 15.6%, $n = 6$) (Fig. 7b, top). Transcriptome analysis of OHBBN bladder tumors also demonstrated heightened expression of EMT and claudin-associated genes. Thus, we tested whether lineage plasticity could exist for EMT-marker-positive cells by isolating and implanting GFP$^+$Epcam$^{low}$CD49f$^{low}$ (G3) cell populations to FVB/NJ mice. Grafts that formed palpable masses of sufficient size for FACS analysis (50,000 cell implants, 9/10 grafts) were then assessed for Epcam and CD49f expression. When gating on GFP$^{pos}$CD45$^{neg}$ cell populations, we observed redistribution to all four quadrants (G1 = 10%, G2 = 12.0%, G3 = 42.3%, and G4 = 30.0%, $n = 5$) (Fig. 7b, bottom). These data show that isolated basal and EMT-like tumor subpopulations are capable of in-vivo plasticity in surface antigen expression.

**Lineage plasticity can occur in human bladder cancer.** To determine whether plasticity in surface antigen expression can occur in human bladder tumors, we established patient-derived xenograft (PDX) models from patients with muscle-invasive bladder cancer (MIBC, $n = 3$) and characterized these as being CD49f$^{high}$;Epcam$^{low}$ in expression. Using an in-vivo approach, tumor cells were isolated using Epcam$^{low}$;CD49f$^{low}$ and Epcam$^{low}$;CD49f$^{high}$ flow cytometry gates and implanted subcutaneously to athymic Fox1a nude mice ($n = 3$) (passage 0, first implant, >50,000 cells). Eight to 12 weeks later, we observed marked tumor formation from CD49f$^{low}$ implant populations, which, upon flow cytometry analysis, showed increased expression in Epcam-low–CD49f-low tumor populations but also marked increases in CD49f-high expression and Epcam-high populations. Examples of such flow cytometry analysis and pathologies are shown in tumors 1 and 2 derived from donor tumor 15934 (Fig. 7c). These data provide evidence that basal and EMT-like cells can undergo lineage plasticity in transplantable models of human, muscle-invasive bladder tumors.

## Discussion

Our studies have demonstrated significant heterogeneity in the urothelia of human tumor samples[9]. Previous analysis from Lund used immunohistochemistry strategies to also support the presence of lineage heterogeneity[4,5] including the identification of mesenchymal-like and small cell-like phenotypes[6]. Despite these findings, there has been emphasis to use bulk transcriptomic data to classify molecular subtypes of urothelial cancers[1,2] as a means to predict tumor progression kinetics, clinical treatment response, and overall survival. Such bulk tumor stratification assumes that each tumor has a predominant static signature sufficient to predict response to therapy and prognosis. Despite this knowledge gap, studies to date have not addressed whether subtyping is static or whether it can be considered a dynamic classification.

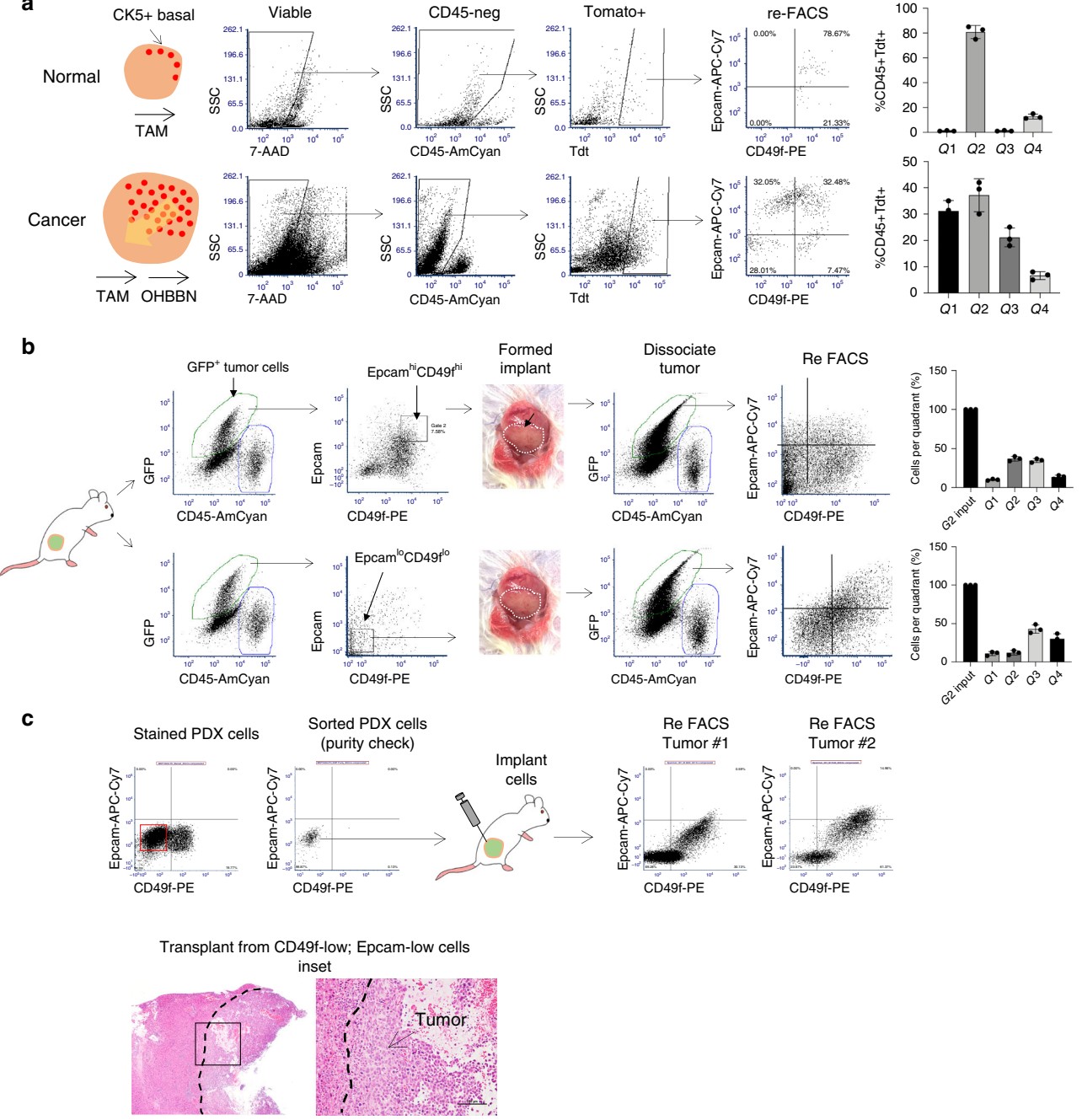

**Fig. 7 Lineage plasticity of basal and mesenchymal tumor cell populations isolated from mouse and human bladder tumors. a** Cytokeratin 5 tomato reporter (*KRT5^ERT2^-LSL*) mice showing that CK5-positive basal cells are enriched for CD49f expression in normal bladders (top). Subsequent to OHBBN-induced carcinogenesis, Tdt cells show heterogeneity in CD49f and Epcam expression (bottom). Right bar graphs, distribution of Tdt-positive;Epcam-CD49f-positive cells in normal and OHBBN-induced bladder cancer. **b** In-vivo transplantation of CD49f-high and CD49f-low tumor populations. Sorted and transplanted basal (GFP⁺Epcam^high^CD49f^high^) cells or mesenchymal cell populations (GFP⁺Epcam^low^CD49f^low^) were implanted to the bladder muscle wall of FVB mice and evaluated for tumor formation. Formed tumors were dissociated to single cells and assessed for redistribution of Epcam-CD49f expression according to the defined quadrants (Q1, Q2, Q3, and Q4) after progression. **c** CD49f-low cells were isolated by flow cytometry from an early (P0) passage PDX model of MIBC (tumor 15934) followed by transplantation to immune compromised mice. Expression analysis of derived tumors under the same flow cytometry settings, shows marked redistribution and increased expression in CD49f ($n = 5$). Lower, representative histology of resulting tumor outgrowths derived from CD49f^low^Epcam^low^ cells (bar = 100 μM). Individual data points are represented by circles, the error bars represent mean ± SD of the three independent replicates. Each experiment was performed at least six times.

In this study we: (1) studied lineage heterogeneity using single cell and bulk transcriptome profiling of bladder tumors, (2) identified antigen combinations which facilitated the isolation of lineage-specific bladder cells, (3) developed a transplantable, immune competent mouse model amenable for lineage progression analysis in primary and metastatic organ sites, (4) determined that cellular expression of lineage markers is dynamic during tumor progression, and (5) demonstrated that human primary and metastatic tumors can be multi-lineage in composition. These data have implications for contextualizing the

current approaches of molecular classification of urothelial cancer, understanding the relative tumor-initiating potential of lineage subpopulations and possibly reconciling the variability in clinical outcomes observed with current lineage subtyping approaches.

Previously reported sequencing analysis of human bladder cancers has identified five mRNA subtypes including luminal–papillary, luminal–immune, luminal, basal–squamous, and neuronal[2]. Our study argues that although these signatures may represent the predominating mRNA expression pattern in a tumor, they do not consider the presence of other cell lineages and, by extension, the possibility that subtyping can change either during progression or in response to treatment. The ability of isolated cell lineages to reconstitute a heterogeneous tumor composed of multiple lineages suggests that tumors may strive for equilibrium or balance in cellular composition. A lineage equilibrium may allow the tumor to more readily adapt to various forms of stress including treatment, progression, metastasis, or even surgical removal of the bulk tumor. We identified cells that simultaneously express multiple lineages, suggesting the presence of cellular transition states occurring between single and multi-lineage marker-positive states. Indeed, recent studies observing that human organoids high in basal lineage marker expression can become more luminal upon transplantation to xenograft environments also implies the presence of transitory cellular states[8]. Data from other reports suggest that subtype switching may occur subsequent to neoadjuvant chemotherapy in bladder cancer[1,10]. Finally, lineage plasticity has also been observed in mouse and human models of prostate cancer, primarily in the setting of cancer treatment[11,12].

Investigators have suggested the presence of a "systematic disagreement" in subtype classification between bulk mRNA and immunohistochemistry profiling[5] based on the supposition that OHBBN-induced bladder tumors are homogenous for basal protein marker expression. However, our single-cell transcriptome analysis revealed the presence of populations within mouse tumors containing cells with significant overlap in expression of basal, luminal, and EMT-claudin gene expression. Our data identify the presence of cells having uni-, bi-, or even tri-lineage gene expression, including cells having coordinately high expression in markers of different subtypes. Further studies will be required to functionally discriminate cell populations with Claudin-high and -mid vs. Claudin-low expression. Although distinct transcriptional populations have been assigned to EMT-Claudin-low cells (a subset of basal-like tumors)[4,13], it is unclear how such cells relate to those Claudin-low cells detected at the immuno-histological level. Our immuno staining studies detected cells with mid and low expression of Claudins 3, 4, and 7 co-expressing basal, luminal, and mesenchymal markers. Immunostaining analysis showed that tumor cells could be single-, double-, or triple-lineage positive. Although speculative, these data suggest that individual cells or populations expressing more than one lineage marker, may be more susceptible to lineage switching than cells expressing a single predominant subtype marker. Our analysis suggest that basal and EMT-like cells are more plastic than luminal cells—a finding requiring further exploration. In-vitro data demonstrate that lineage plasticity can occur in a cell autonomous manner. However, microenvironment components (including immune cells and cancer associated fibroblasts (CAFs)), which are typically excluded in conventional sphere or organoids assays, may also regulate plasticity between basal, luminal, mesenchymal, and potential other lineages or subtypes. Future studies may consider how these factors influence in-vivo lineage plasticity particularly in the context of therapy.

Addressing the potential for lineage plasticity in our model required the identification of surface antigens that could distinguish mRNA subtypes within tumor cell populations. We identified CD49f to be highly expressed in OHBBN-induced bladder cancers and, thus, tested whether the CD49f-Epcam marker combination may provide a method to discern lineage heterogeneity within the same tumor as previously hypothesized[6]. Other studies using PDX models of bladder cancer have demonstrated that CD49f, in conjunction with surface antigens such as CD44, CK20, CD47[14], and CD90[15], discriminate poorly differentiated (basal) from differentiated bladder cancer cells. Our studies in prostate[16,17] and those by others[18,19] demonstrated that CD49f enriched tumor cells or transduced normal prostatic epithelia are capable of basal cell-initiated tumor formation and cellular differentiation. The importance of basal cells for tumor initiation and progression can be appreciated from the fact that p63-positive cells differentiate into all epithelial lineages of the bladder[20]. Moreover, p63 loss leads to abnormal bladder development and function. Thus, although transformation of a basal cell(s) may have the potential to propagate into other cell lineages, during the evolution of bladder cancer progression, it remains to be determined under what conditions (genetic alternations and pathological) other lineages including luminal and mesenchymal cells can do the same. Our analysis determined that luminal cell implants were less capable of tumor initiation and lineage plasticity—a finding that requires additional mechanistic understanding. Indeed, the recent identification of cells with a neuroendocrine signature in human bladder tumors suggests that certain cells may have an even broader scope for cellular and lineage plasticity than previous appreciated.

An interesting finding from our study was that enriched luminal and mesenchymal cells could efficiently revert to basal marker high cell populations subsequent to transplantation and progression. These findings are compatible with studies showing that CD49f expression can demark breast cancer cells capable of conversion between epithelial and mesenchymal-like states[21]. These data suggest the possibility of a reverse EMT, whereby cancer cells with mesenchymal qualities are capable of plasticity and, under favorable primary tumor growth conditions, switch to other more proliferative cell lineages similar as studied in other tumor types[22]. Conversely, during chemotherapy, the accumulation of mesenchymal cells may serve as a mechanism for acquired resistance[23] potentially occurring by lineage switching from more chemo sensitive cell types. Such data are also in line with studies observing that taxane exposure can promote increased mesenchymal content[24] during treatment of castration resistant prostate cancer[25] and occurrence of chemo resistance in bladder cancer[26–28].

If our preclinical studies are predictive of how lineage plasticity may occur during human tumor progression, then several clinical implications can be identified. Tumor progression and treatment response may be dependent upon the relative contributions of various cellular subtypes that comprise an individual tumor. The potentially dynamic nature of molecular subtypes occurring during cancer therapy may justify repeat tumor sampling with serial lines of treatment being required to optimize the use of gene expression data to guide "precision medicine". Finally, new conceptual approaches of treatments to overcome heterogeneity and plasticity may be applied including rapid cycling of non-cross resistant systemic therapies as has recently been proposed for prostate cancer[29].

## Methods

**Genetically engineered mouse models.** *Tg(CAG-luc-eGFP)* FVB/NJ bladder cancer model: Mice were obtained from JAX (008450) and bred on a homozygous background. To develop bladder-specific cancer, mice of at least 6 weeks of age were treated with 0.1 % OHBBN (TCI, B0938) for 14 weeks and allowed to progress for an additional 4–6 weeks. Upon detection of palpable masses or hematuria, primary bladder tumors were resected and transplanted subcutaneously to FVB/NJ (JAX, 001800) male or female mice constituting a passage 0 or "P0"

tumor. Successful P0 tumor outgrowths were then banked as frozen stocks and used for experimental studies as P1 implants.

$KRT5^{ERT2}-R26^{LSL}$-Tdt bladder cancer model: Krt5-ERT2 mice were obtained from Pierre Chambon (IGBMC) and were crossed with R26$^{LSL-Tomato}$ mice (JAX, 007909) to generate tomato reporter mice under the control of the Ck5 promoter. $KRT5^{ERT2}-R26^{LSL}$-Tdt were treated with OHBBN for 14 weeks followed by resection of primary tumors and S.Q. transplantation to athymic Nude Fox1a mice (Envigo). To activate the Krt5-cre, mice were treated with TAM (4 daily consecutive doses, 75 mg/kg, i.p.) and evaluated for reporter expression.

Patient-derived xenograft models: PDX models of bladder cancer were derived using fresh surgical specimens obtained by cystectomy. All patient tissues were MIBCs and implanted subcutaneously in athymic Nude Fox1a mice (Envigo) in a mixture of 80% Cultrex-20% media.

**Mouse ethics and housing conditions**. All mouse experimentation was conducted according to an approved Institutional Animal Care and Use Committee (IACUC) protocol (LA13-00060). Mice were housed in an Association for Assessment and Accreditation of Laboratory Animal Care International (AAALAC)-approved vivarium including hepa filtration, ad libitum supply of food and water, and daily health monitoring.

**Human ethics and tissues samples**. Formalin-fixed paraffin-embedded (FFPE) MIBCs were obtained from either transuretheral resection or radical cystectomy specimens. These samples were from patients, consented under an IRB-approved biorepository protocol, who were chemo-naive ($n = 6$) and chemo-resistant ($n = 6$). FFPE samples from patients harboring metastatic samples were also obtained ($n = 17$). All human samples were acquired under an approved Institutional Review Board (IRB) protocol (IRB 10-1180).

**Ivis imaging**. Mice were injected i.p. with 150 mg of luciferin/kg of body weight 10–15 min before in-vivo imaging and subsequently analyzed using an IVIS Spectrum.

**Surgical tumor cell implantations**. To generate tumor engraftments in the bladder or liver, single-cell dissociates were generated from OHBBN-induced bladder tumors using combined enzymatic digestion (2–3 h at 37C, type I collagenase, Gibco) and mechanical dissociation (10 slow passes with a 21 G needle). Dissociates were washed in media, counted and resuspended in an 80% Cultrex-20% media solution. For each injection, 2.5–5 µl of a Cultrex-media solution was used containing as few as 2,000 and up to 50,000 cells. Using the desired volume of Cultrex-media, cells were loaded into a microinjection needle (Hamilton, 80408) and incubated at room temperature for 90–120 s, to reach the desired injection consistency. The cell composite was then injected to either the bladder muscle wall or the medial lobe of the liver using FVB/NJ mice (JAX, 001800). Successful tumor cell implants typically produced visible tumor masses with 15 days post injection.

Variables affecting the growth kinetics of tumor implants include aggressiveness of the primary tumor line, passage number of donor subcutaneous tumor and health of the injected tumor cells. To tag cells with eGFP or luciferase expression, donor tumor cells were obtained from OHBBN-treated *Tg(CAG-luc-eGFP)* mice (JAX, 008450) or from OHBBN-treated FVB/NJ mice (JAX, 001800) infected with HIV-CMV-eGFP lentivirus (Viral Vector Core, University of Iowa).

**Flow cytometry, immunohistochemistry, and antibodies**. All flow cytometry analysis was completed on a BD LSR Fortessa using BD FACS Diva software. Cell sorting was completing using a BD FACS Aria II model and BD FACS Diva Software. For immunohistochemistry, tissue specimens were procured through the Oncological Sciences Histology (Icahn School of Medicine at Mount Sinai). Immunostaining was conducted according to conventional procedures. Antibodies used for immunocytochemistry and immunohistochemistry are listed in Table 3.

**Single-cell transcriptome analysis**. Single-cell RNA-seq data were preprocessed with the scater[30] and normalized by scran[31]. Data integration, unsupervised cell clustering, and differential expression analysis were carried out by the Seurat v3.0[32]. Reference-based cell type annotation was generated by SingleR[33]. Cells with >3 median absolute deviation were removed as outliers. Cells with less than 400 genes or 1000 UMIs, or >15% of mitochondria genes were filtered out from the analysis. Altogether, the filtered data contained 27,998 cells and 24,421 genes from 6 samples. Cell-specific biases were normalized using sctransform normalization in Seurat[34]. The top 3000 highly variable genes were selected using the expression and dispersion (variance/mean) of genes, followed by the reference-based data integration. The first 105 prinicpal component analysis (PCA) results were chosen to generate dimensional tSNE plots, Uniform Manifold Approximation and Projection plots, and cell clustering by a shared nearest-neighbor modularity optimization-based clustering algorithm. Cell types were manually identified by marker genes[2] and confirmed by SingleR (Single-cell Recognition) package using 358 mouse RNA-seq[33]. Differential expression analysis was calculated between either two tested groups or one group vs. all other groups, based on the MAST (Model-based Analysis of Single Cell Transcriptomics)[35]. We consider gene with a Bonferroni-corrected q-value < 0.05 and log2(fold change) > 0.1. The raw results from Seurat differential expression analysis are based on the natural log. They are all converted to log2 in our comparison. Cells were gated with Ck5, Ck8, and Krt14 expression using Feature Scatter function to visualize gene-to-gene relationships.

**Microarray data set and data analysis**. We obtained publicly available Illumina mouse-6 v1.0 expression beadchip[36]. The data set comprised of ten samples: five samples with OHBBN-induced mouse bladder tumors and five samples with normal bladders. This data was downloaded as preprocessed Series Matrix data for 46,006 probe sets. The analysis was done in R statistical environment. Probes were collapsed to gene level after taking the mean expression of the probes, using the

---

**Table 3 Antibodies used for flow cytometry and immunohistochemistry analysis.**

**Antibodies for flow cytometry**

| Antibody name | Company | Catalog | Clone | Fluorophore | Dilution | Validation |
|---|---|---|---|---|---|---|
| CD45 (Ms) | Biolegend | 103138 | 30-F11 | BV510 | 1/500 | IgG, FMO |
| CD24 (Ms) | Biolegend | 101822 | M1/69 | PE/Cy7 | 1/500 | IgG, FMO |
| CD44 (Ms/H) | Biolegend | 103018 | IM7 | Alexa 647 | 1/500 | IgG, FMO |
| CD49f (Ms/H) | Biolegend | 313611 | GoH3 | PE | 1/250 | IgG, FMO |
| CD49f (Ms/H) | Biolegend | 313610 | GoH3 | Alexa 647 | 1/250 | IgG, FMO |
| Epcam (Ms) | Biolegend | 118208 | G8.8 | FITC | 1/500 | IgG, FMO |
| Epcam (Ms) | Biolegend | 118218 | G8.8 | APC/cy7 | 1/500 | IgG, FMO |
| Epcam (Ms) | Biolegend | 118216 | G8.8 | PE/Cy7 | 1/500 | IgG, FMO |
| Viability-7aad | Biolegend | 420403 | | PerCP | 1/100 | No stain |

**Antibodies for Immunohistochemistry**

| Antibody name | Company | Catalog | Dilution | Validation |
|---|---|---|---|---|
| CK5 | Biolegend | 905501 | 1/2000 | FMO |
| CK8 | Biolegend | 904801 | 1/1000 | FMO |
| CK8/18 | Abcam | Ab194130 | 1/50 | FMO |
| Epcam | Abcam | ab71916 | 1/100 | FMO |
| Vim | Abcam | MA1-37027 | 1/100 | IgG, FMO |
| SMA | Invitrogen | MA1-37027 | 1/100 | IgG, FMO |
| Ki67 | Thermoscientific | RM-9106 | 1/200 | FMO |
| Claudin 3 | Abcam | ab15102 | 1/100 | FMO |
| Claudin 4 | Abcam | ab15104 | 1/200 | FMO |
| Claudin 7 | Abcam | Ab27487 | 1/200 | FMO |

gene annotations for Illumina mouse-6 v1 beadchip. We used this data to do clustering using subtypes (basal, luminal, mesenchymal, and neuronal) described in Robertson et al.[2]. The Complex Heat Maps Bioconductor package was used for heat map visualization.

**Flow cytometry sample preparation**. Flow cytometry was conducted on OHBBN-induced mouse and human bladder tumors. For both, tissues were mechanically minced with a razor blade followed by enzymatic digestion using collagenase, rotating for 2–4 h at 37 °C. During this time, digestion mixtures were passed 2× through a 21.5 G needle and syringe. After digestion, the mixture was passed through a 100 µM filter and washed in PBS using 2× 5 min cycles of centrifugation at 900 r.p. m. Pellets were then resuspended in FACS staining buffer (PBS, 1% BSA, 0.05% P/S) to produce single-cell mixtures ready for primary antibody staining (Table 3).

**Quantitative PCR**. q-PCR analysis was done by conventional protocols. In brief, sorted cells were collected directly into Trizol LS (Invitrogen) followed by purification by RNAeasy mini kits (Qiagen). RNA was then subject to cDNA synthesis and analysis using sequence specific primers. Primer sequences used for q-PCR are shown in Supplementary Table 2.

**Reporting summary**. Further information on research design is available in the Nature Research Reporting Summary linked to this article.

## Data availability

The Human bladder data referenced during the study are available in a public repository from the https://weillcornell.shinyapps.io/Human_BladderCancer/. The mouse bladder data referenced during the study are available in a public repository from https://weillcornell.shinyapps.io/Mouse_BladderCancer/. The scRNA-seq data have been deposited in the GEO database under the accession code GSE146137 [https://www.ncbi.nlm.nih.gov/geo/query/acc.cgi?acc=GSE146137]. The differential analysis results are provided as a Source Data file at https://github.com/nyuhuyang/scRNAseq-BladderCancer/tree/master/DEGs. All the other data supporting the findings of this study are available within the article and its Supplementary Information files and from the corresponding author upon reasonable request. A reporting summary for this article is available as a Supplementary Information file.

## Code availability

The scripts used for analysis and figure generation are available at https://github.com/nyuhuyang/scRNAseq-BladderCancer.

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

## Acknowledgements

We thank Joy Park, Alison Park, Jade Tapia, and Aileen Li for general assistance in this project. We thank Michael Donovan (Icahn School of Medicine) for providing human samples of metastatic bladder cancer. We acknowledge the Human Immune Monitoring Center at Mount Sinai.

## Author contributions

J.P. helped write the manuscript and provided patient bladder tumor samples. J.D., H.A., G.B., and D.M. conducted the experiments. Y.H., R.B., and O.E. completed

bioinformatics analysis. K.B., M.G., and B.F. provided valuable experimental advice and helped write the manuscript. D.M. wrote the manuscript.

## Competing interests

The authors declare no competing interests.
