## [Peer Review File · Nature Communications]

Reviewers' comments:

Reviewer #1 (Expertise: Bladder cancer mouse model, Remarks to the Author):

This manuscript by Anastos et al seek to dissect intratumoral heterogeneity and plasticity within urothelial carcinomas, using a transplantable line of BBN-induced murine bladder cancer as a model. They claim the existence of multiple histologic morphologies within a single tumor, and identified subpopulations are capable of equal tumor initiation and plasticity. However, the current experimental designs and certain technical concerns may not back up these strong claims:

1) Fig 2b suggest the existence of basal, luminal and mesenchymal mixed morphologies within a single transplantable tumor allograft. If true, this could be potentially interesting.

- It is advisable to perform IF co-staining of various markers, using low mag to overview the entire tumor, showing epi cells are all GFP+ and zoom into high mag to indicate relative localizations and contributions of different morphologies within the same tumor. Fig 2 & 3 are claiming a generalization across many BBN/allograft tumors, a table with n number clarifying the total number of transplantable tumors analyzed, versus various morphology is essential.

- Can they show patient tumors including all three morphologies (within a single tumor) to demonstrate clinical relevance?

2) Technically concerning that many FACS plots are inconsistent, or indicating heterogeneity may be pre-existing across tumors they were analyzing.

- For instance, Fig 4a and Fig S4c are completely different. Fig 4a showed G1-G4 populations while G1 disappeared in Fig S4c with another new CD49fmedium population, raising concerns regarding compensation or further tumor heterogeneity.

- Fig 4c(lower panel) and Fig S6b are completely different. Fig 4c showed G1, G2 indicating CK5+ cells primarily give rise to basal cells, while Fig S6b indicates otherwise i.e. complete lack of G1. What are they trying to show?

- According to their compensation and GFP+ gating, almost all CD45+ cells are GFP+? Please explain. There are no GFP-negative population which should comprise of stromal/CD45 immune infiltration, raising further technical concern of their "mesenchymal derived cells".

3) If one claims that all G1-4 are capable of tumor initiation, limiting dilution experiment injecting high to low cell number is required for comparison side-by-side in a table, across tumor allografts.

No indication what cell numbers was injected for CK5EpcamhighCD49high or EpcamlowCD49low population in Fig 4 and 5 respectively, and how many allografts were used.

4) Is there any evidence that basal and luminal lineage should not cross, in a normal or neoplastic context? Cellular differentiation is not surprising. It would be interesting if they can demonstrate in a certain biological context that lineage switch is clinically relevant.

Minor points:

1) Fig 3c, their CK8+/CK5+ claim is concerning, CK5 should not be a nuclear stain.

2) They made claim for existence of heterogeneity in BBN tumors, then in Fig 2 they analyze public available dataset of BBN tumors, instead of their own Tg(CAG-luc-eGFP) tumors that were used to generate tumor allografts?

3) Fig. S6 uses culture conditions that will skew toward stemness to grow a mix population into basal biased KRT5Cre.Tdt organoids, and then transplant a mixed population into an in vivo microenvironment that enable growth of all populations. How would that strengthen their argument?

4) Unclear how Fig 6 will strengthen their point by gene expression? Gene expression does not indicate subpopulations? It is unclear why they stain for other patients (Fig S7 & S8) but not those in Fig 6?

Reviewer #2 (Expertise: MIBC genomics, therapy, Remarks to the Author):

Recent studies performed by TCGA and academic centers in North America and Europe identified “intrinsic” basal and luminal molecular subtypes and have linked them to patient prognoses and responses to conventional and targeted therapies. However, these studies mostly involved whole transcriptome profiling of bulk tumors, leaving open the question of whether or not primary tumors display subtype-related molecular heterogeneity. The authors of the present study used carcinogen-induced bladder cancers from mice and primary bladder cancers from patients to address this question. They identified basal, luminal, and mesenchymal cell subsets in bulk tumors and identified surface markers that allowed for their isolation. They then showed that these isolated populations displayed significant plasticity such that they could recreate the other cellular populations in the bulk tumors. While these experiments do directly address the question, there are several major concerns, mostly related to innovation and summarized in a point-by-point manner below.

1. Formal assignment of tumors to the various molecular subtypes identified previously (k = 2/UNC to k = 6/updated Lund classification) requires the use of tens to thousands of genes and formal bioinformatics tools. The fact that the authors did not use such methods represents a major limitation of the study. Using single markers to identify basal versus luminal versus claudin-low tumors may be feasible (indeed, IHC studies have concluded this), but for a paper of this impact, rendering conclusions based on single markers (or even handfuls of markers) is premature.
2. Related to the point above, the luminal subtypes can be subdivided into tumors that are mostly devoid of basal biomarkers (the Lund “genomically unstable” subtype) and tumors that co-express basal and luminal biomarkers (the Lund “urothelial A” subtype). Immunochemical analyses of the uroA tumors by the Lund group demonstrated that the areas of the tumors that were in contact with the basement membrane expressed high levels of basal biomarkers. Therefore, the observation that luminal tumors can contain basal cells is not novel. Furthermore, because the group did not use the Lund classifier, whether they would have observed “basal” cells in the GU tumors cannot be determined. (A similar phenomenon has been observed by Andy Ewald’s group at tumor edges during collective invasion of luminal breast cancers.)
3. Regarding subtype “plasticity”, the recent paper by Michael Shen’s group describing bladder cancer organoids already established that this can occur. About two thirds of their “luminal” organoids became “basal” with passage ex vivo; they then switched back to more “luminal” phenotypes when they were grown as orthotopic xenografts. Again, this undermines the novelty of the paper.
4. The authors make a point of emphasizing the novelty of their mouse model. While it is innovative in its details, it is an adaptation of a very widely-used carcinogen-induced model, and this innovation is over-stated.
5. Related to the above, previous studies demonstrated that bulk muscle-invasive BBN-induced bladder cancers are almost all basal. Did the authors observe the same thing?
6. The conclusions in this study would be dramatically strengthened by the inclusion of single-cell RNAseq data.

Reviewer #1

This manuscript by Anastos et al seek to dissect intratumoral heterogeneity and plasticity within urothelial carcinomas, using a transplantable line of BBN-induced murine bladder cancer as a model. They claim the existence of multiple histologic morphologies within a single tumor, and identified subpopulations are capable of equal tumor initiation and plasticity. However, the current experimental designs and certain technical concerns may not back up these strong claims:

1) Fig 2b suggest the existence of basal, luminal and mesenchymal mixed morphologies within a single transplantable tumor allograft. If true, this could be potentially interesting.

- It is advisable to perform IF co-staining of various markers, using low mag to overview the entire tumor, showing epi cells are all GFP+ and zoom into high mag to indicate relative localizations and contributions of different morphologies within the same tumor.

The reviewer has made a good suggestion. Therefore, we have obtained the appropriate reagents and conducted low and high magnification co-IF of mouse derived allografts derived from OHBBN induced primary bladder tumors. Images include GFP stains counterstained with basal (CK5, p63), luminal (CK8) and mesenchymal lineage markers (Vim, Sma). These data are found in Supplemental Figure 10.

2) Technically concerning that many FACS plots are inconsistent, or indicating heterogeneity may be pre-existing across tumors they were analyzing.

The reviewer points out that tumors used in this study may have heterogeneity.

In general, even genetic engineered models (GEM) of cancer (non carcinogen initiated) show heterogeneity amongst litter mate controls, thus, demanding the use of replicates and multiple mice for analysis. The OHBBN induced bladder cancer model is carcinogen induced and therefore (like human disease) is subject to variable mutational landscapes for each primary tumor and derived allograft. As a result, yes, individual tumors from this model may progress with slightly different kinetics, degrees of invasiveness and immune cell infiltration.

However, to address the reviewers concern we have made efforts on several levels to improve our flow cytometry analysis including: (1) increased numbers of CD49f-Epcam FACS analysis using expanded tumor cohorts of primary and subcutaneous tumors (MDAs) from OHBBN treated mice (**Fig S5A, B**), (2) inclusion of representative compensation controls (IgG, FMO) (**Fig S6B**), (3) and quantitation of expression analysis for mouse tumors.

We have also evaluated expression of CD49f-Epcam expression in primary, human bladder tumors (from radical cystectomies (**Fig. S6A**) including distribution analysis.

Thus, despite some inherent (biological) heterogeneity in primary mouse and human tumors, we have taken the appropriate biological and technical controls to hopefully address the reviewers concern.

- For instance, Fig 4a and Fig S4c are completely different. Fig 4a showed G1-G4 populations while G1 disappeared in Fig S4c with another new CD49f medium population, raising concerns regarding compensation or further tumor heterogeneity.

Agreed – Figure 4c (now Figure 6b) has been replaced with additional data including mice having the full course (14 wks.) of OHBBN treatment and paired, non-BBN treated controls. Specifically, Figure 6b (top) shows FACS analysis of bladder cells pulsed mice with TAM to induce tomato expression (tdt) in Ck5 positive cells (a basal marker). In these normal urothelia, TAM pulsed KRT5-tomato bladders form (differentiate to) CD49f high and mid expressing cells (Q2 and Q4). Conversely, mice pulsed to form KRT5-tdt positive cells and subsequently treated with OHBBN yielded cells in all quadrants including Q1, Q2, Q3 and Q4 upon analysis for CD49f-Epcam. Since CK5 and CD49f are both basal markers, these data are meant to corroborate that CD49f positive cells can form multiple lineages in the context of cancer.

- Fig 4c(lower panel) and Fig S6b are completely different. Fig 4c showed G1, G2 indicating CK5+ cells primarily give rise to basal cells, while Fig S6b indicates otherwise i.e. complete lack of G1. What are they trying to show?

See above text for response.

- According to their compensation and GFP+ gating, almost all CD45+ cells are GFP+? Please explain. There are no GFP-negative population which should comprise of stromal/CD45 immune infiltration, raising further technical concern of their “mesenchymal derived cells”.

The CD45+ immune cells are not GFP+ as they are low in the GFP axis. However, we recognize that confusion arising from the level of GFP expression. Using more appropriate compensation controls (GFP vs FITC) we have improved these plots markedly which now show distinct GFP-high-CD45-low versus GFP-low-CD45-high populations.

3) If one claims that all G1-4 are capable of tumor initiation, limiting dilution experiment injecting high to low cell number is required for comparison side-by-side in a table, across tumor allografts. No indication what cell numbers was injected for CK5EpcamhighCD49high or EpcamlowCD49low population in Fig 4 and 5 respectively, and how many allografts were used.

The reviewer has requested transplant studies using varying numbers of cells which, through considerable efforts, we have now completed. Cell numbers are now included in the results section and figure legends.

For this we have taken two approaches including *in vitro* and *in vivo* limiting dilution studies. For *in vitro*, we have conducted a comparative evaluation of organoid formation of lineages derived from OHBBN induced primary bladder tumor cells. Cell populations from primary tumors (n=5) were harvested and sorted based on CD49f and Epcam expression. Sorted cells were plated in different densities (1, 2 5, or 50 x 10³ cells per 20 µl Cultrex pellet) followed by counting of organoids per pellet 7-8 days after plating. Average organoid counts per pellet are shown in bar graphs corresponding to the respective collected cell populations (**Fig. S9**).

For *in vivo* limiting dilutions experiments populations were isolated from OHBBN induced bladder tumors (primary, first passage MDAs). Cell populations based on CD49f and Epcam expression were sorted and injected directly to the bladder of recipient FVB mice and evaluated to 8-12 weeks later for formed tumor mass in recipient bladders. The efficiencies of tumor formation are shown in table form including absolute and percentage values where NG = no graft formation, NG* = no solid graft formation but viable cells detected with hemocytometer (**Fig. S9**).

We would like to point out that when conducting these studies using a SQ approach, tumor initiation often required at least 50K cells making limiting dilution studies difficult. Using our established transplant, orthotopic approaches (to bladder wall or liver), we observed greater sensitivity in detecting tumor initiation using lower cell numbers. With these *in vitro* and *in vivo* strategies we showed that, while all populations can form outgrowths, populations G2 and G3 are by far the most efficient at initiation.

4) Is there any evidence that basal and luminal lineage should not cross, in a normal or neoplastic context? Cellular differentiation is not surprising. It would be interesting if they can demonstrate in a certain biological context that lineage switch is clinically relevant.

The reviewer poses a challenging question. Lineage differentiation and transdifferentiation likely occurs both in normal and oncogenic environments. Moreover, lineage switching likely also depends upon the cell and organ types considered.

In terms of clinical relevance, this may be related to the response to chemotherapy in which case reports have associated molecular subtypes to treatment with cisplatin chemotherapy (PMIDs: 24525232, 28390739, 26343003). Recent works by Peter Black's group has demonstrated plasticity in the biological response to neoadjuvant chemotherapy in muscle invasive bladder cancer (Seiler, Clinical Cancer Research, 2018). Moreover, this study has also suggested that certain subtypes may not be readily applicable to post chemotherapy bladder cancer.

Thus, if once considers treatment and biological response, key questions for the field will be to dissect how (1) therapies change subtypes, (2) whether adaptive plasticity in subtypes constitute mechanisms of resistance and (3) whether such plasticity is reversible.

Minor points:

1) Fig 3c, their CK8+/CK5+ claim is concerning, CK5 should not be a nuclear stain.

No - Figure 3C (left panel) does not show nuclear CK5, rather most of the CK5 staining is cytoplasmic as expected (green, left panel). Conversely, Ki67 staining (red) is mostly in the nuclear in the same panel. We also validate our CK5 staining conditions in Fig S10, S11 at low and high mag expression patterns. Sometimes, if the cell is in a different plane of focus, there may be an impression of nuclear staining. We will note this in the text.

2) They made claim for existence of heterogeneity in BBN tumors, then in Fig 2 they analyze public available dataset of BBN tumors, instead of their own Tg(CAG-luc-eGFP) tumors that were used to generate tumor allografts?

Yes, this is a fair point. We have now included single cell transcriptome analysis on our primary mouse and human models. The publicly available data previously shown set now serves to corroborate our data analysis using a different methodology.

3) Fig. S6 uses culture conditions that will skew toward stemness to grow a mix population into basal biased KRT5Cre. Tdt organoids, and then transplant a mixed population into an in vivo microenvironment that enable growth of all populations. How would that strengthen their argument?

The reviewer is correct and this cell culture data has now been removed. The original point was to show that basal enriched organoids can differentiate in vivo to form tumors with a greater content of luminal cells.

4) Unclear how Fig 6 will strengthen their point by gene expression? Gene expression does not indicate subpopulations? It is unclear why they stain for other patients (Fig S7 & S8) but not those in Fig 6?

The reviewer is correct and this data has been removed. Single cell transcriptome analysis for human tumors has been included.

Reviewer #2 (Expertise: MIBC genomics, therapy, Remarks to the Author):

Recent studies performed by TCGA and academic centers in North America and Europe identified “intrinsic” basal and luminal molecular subtypes and have linked them to patient prognoses and responses to conventional and targeted therapies. However, these studies mostly involved whole transcriptome profiling of bulk tumors, leaving open the question of whether or not primary tumors display subtype-related molecular heterogeneity. The authors of the present study used carcinogen-induced bladder cancers from mice and primary bladder cancers from patients to address this question. They identified basal, luminal, and mesenchymal cell subsets in bulk tumors and identified surface markers that allowed for their isolation. They then showed that these isolated populations displayed significant plasticity such that they could recreate the other cellular populations in the bulk tumors. While these experiments do directly address the question, there are several major concerns, mostly related to innovation and summarized in a point-by-point manner below.

While challenging - the reviewer's point is very well appreciated. We have, thus, made significant efforts to complete single cell transcriptomic analysis as a means to assess lineage heterogeneity (Fig. 1, 2; Fig. S1, S3). As a result, we view the paper to be significantly strengthened by the resulting innovative data obtained from mouse and human data sets which would not easily be achievable using conventional bulk transcriptome analysis. Innovation has also been enhanced through the use of refined transplant assays showing the marked potential for lineage plasticity between different sub populations defined by CD49f and Epcam expression both in mouse and human MIBCs.

1. Formal assignment of tumors to the various molecular subtypes identified previously (k = 2/UNC to k = 6/updated Lund classification) requires the use of tens to thousands of genes and formal bioinformatics tools. The fact that the authors did not use such methods represents a major limitation of the study. Using single markers to identify basal versus luminal versus claudin-low tumors may be feasible (indeed, IHC studies have concluded this), but for a paper of this impact, rendering conclusions based on single markers (or even handfuls of markers) is premature.

In response to the initial query made, our bioinformatics team would like to point out that even though these studies started out with all 20,000 genes, they eventually converged on consensus classifiers which include only tens to hundreds of genes. So, using these “supervised clustering” classifiers that have been already validated to determine subtypes is a reasonable approach. The reviewer may be implying that the limitation of using classifiers that were originally developed for human bladder cancers to classify murine tumors. A recent publication used BASE47 (so just the 47 genes) to classify BBN tumors (which cluster with basal) vs. their transgenic UPPL luminal tumors derived by lineage-specific PTEN/P53 KO from the umbrella cells. Reference: <http://cancerres.aacrjournals.org/content/early/2018/05/18/0008-5472.CAN-18-0173>. We have cited this publication and included the figure below for reference.

2. Related to the point above, the luminal subtypes can be subdivided into tumors that are mostly devoid of basal biomarkers (the Lund “genomically unstable” subtype) and tumors that co-express basal and luminal biomarkers (the Lund “urothelial A” subtype). Immunochemical analyses of the uroA tumors by the Lund group demonstrated that the areas of the tumors that were in contact with the basement membrane expressed high levels of basal biomarkers. Therefore, the observation that luminal tumors can contain basal cells is not novel. Furthermore, because the group did not use the Lund classifier, whether they would have observed “basal” cells in the GU tumors cannot be determined. (A similar phenomenon has been observed by Andy Ewald’s group at tumor edges during collective invasion of luminal breast cancers.)

It is worth noting that previous analysis of heterogeneity from human bladder cancers generally used RNA sequencing and immunohistochemistry from bulk tumors including the UNC and Lund Classifications. To our knowledge, our study is the first to use single cell RNA sequencing analysis to dissect heterogeneity in human bladder cancers.

Our single cell transcriptomic analysis provides new insight that multiple lineages can coexist within single tumors both from mouse and human bladder tumors. These data show that tumors can not only contain basal and luminal populations but can also include EMT cell populations capable of bidirectional differentiation. We argue that our work expands by providing mechanistic insight by exploiting an established animal model to allow lineage tracing of the tumor cells and prove the subtype switching/plasticity. These data is included in **Figures 1, 2; Figures S1, S3**.

3. Regarding subtype “plasticity”, the recent paper by Michael Shen’s group describing bladder cancer organoids already established that this can occur. About two thirds of their “luminal” organoids became “basal” with passage ex vivo; they then switched back to more “luminal” phenotypes when they were grown as orthotopic xenografts. Again, this undermines the novelty of the paper.

We view these studies as complementary. The study of Michael Shen focuses on organoid cultures and the phenomenon of plasticity between in vivo SQ environments and in cell culture. Moreover, Michael Shen’s organoids are not of a “purified” starting populations (i.e. organoid in vitro are not only basal). Thus, similar to Choi et al., Shen et al. show that you have divergence of subtype secondary to external pressures i.e. tumor growth or progression. Our analysis is more specific in that we use defined lineage markers (CD49f-Epcam) to show plasticity between isolated and specific cell lineage types including basal, luminal and EMT in mouse and human. We show plasticity at the cellular level rather than tumor level.

4. The authors make a point of emphasizing the novelty of their mouse model. While it is innovative in its details, it is an adaptation of a very widely-used carcinogen-induced model, and this innovation is over-stated.

Agree – in the revised manuscript we have made less emphasis on model innovation and more emphasis to demonstrate how the model was used to achieve innovative data. We have identified five areas where our adapted modeling is been highly innovative:

1) Conducting limiting dilution transplants. Using conventional SQ transplants we were unable to demonstrate significant differences between independent lineage populations particularly at low input numbers. However, by using surgical orthotopic transplants to the bladder muscle wall or liver, we achieved clear results.

2) Discerning EMT cancer cells from the normal host mesenchymal population. As there are few (if any) markers that can discriminate normal and transformed mesenchymal cells, this model will also lend well for those wishing to discriminate transplanted tumor cells and host immune cells. In conjunction with use of CD49f + Epcam surface antigens we have been able to study the contribution of EMT cells towards cellular plasticity and tumorigenesis.

3) Evaluating the effects of the impact of different tumor immune microenvironments. While outside the scope of this study, we have leveraged this model to develop transplant models to the bladder, liver and bone which may be useful for study the effects of different clinically relevant tumor microenvironments on tumor progression, treatment resistance.

4) Allows for non-invasive bioluminescent imaging of transplants. Progression studies using transplants to the bladder and liver may be strengthened since cancer cells are tagged with the firefly luciferase gene.

5) To our knowledge reporting the first OHBBN derived bladder tumor single cell RNA sequencing data. This data shows individual cells can maintain more than one subtype.

5. Related to the above, previous studies demonstrated that bulk muscle-invasive BBN-induced bladder cancers are almost all basal. Did the authors observe the same thing?

While there is predominant basal signature using “bulk” transcriptome analysis, our bioinformatics analysis also identified several other sub-populations including luminal and EMT. These were corroborated by three methodologies including: (1) immunohistochemistry detection of luminal and EMT markers in primary OHBBN induced primary bladder tumor (**Fig. S10**), (2) use of surface antigen combinations (CD49f-Epcam) used to isolate basal, luminal and mesenchymal cell populations within primary tumors and (3) use of single cell transcriptome analysis to demonstrate that multiple subtypes exist within OHBBN bladder tumors.

Thus, if one examines a cohort comprising several biological replicates (tumors from different mice that were treated with BBN), then the answer is, yes, the primary subtypes signature is “basal”, however, if one is referring all cell types within one OHBBN tumor then the answer is no, as we have detected other lineages secondary to basal within most primary tumors examined.

6. The conclusions in this study would be dramatically strengthened by the inclusion of single-cell RNAseq data.

Agreed. We have devoted significant efforts to completing the single cell transcriptome studies which have dramatically strengthened our study.

Reviewers' comments:

Reviewer #1 (Remarks to the Author):

The authors have put in considerable efforts to provide a significantly improved manuscript, with the incorporation of single cell sequencing data from n=2 BBN-induced mouse tumors and n=2 human MIBCs, much improved FACS compensation and the attempts of new in vivo tumorigenic experiments from FACS-purified populations. These are highly encouraging; however, their use of certain technical terms and concepts within the manuscripts are extremely confusing to the audience and not necessarily supported by their elegantly generated data. In brief, the terms such as lineage tracing, multi-lineage progenitors, cell-of-origin etc are loosely used basing upon molecular data.

1. Intra-tumor heterogeneity was reported in bladder cancer, with functionally distinct tumor subpopulations with molecular features similar to that during normal cellular differentiation [PNAS 2009;106:14016-21]. It is therefore widely accepted that a “molecular subtype” reflects the gene signature of a predominant cell type within a bulk population. These authors take an important step forward by performing single cell sequencing to identify four unique populations of CD45 neg epithelial cancer cells.

1a) Molecular subtype or expression of a single cell does not define a “lineage”, we suggest to remove languages such as bi- or tri-lineage unless they can back up their “lineage tracing” claims with bar-coding type of functional lineage tracing experiments. None of the experiments in Fig 5 are considered “lineage tracing”. 1b) One cannot designate a cell as coexpressing “markers” based on relative mRNA expression, and directly start referring to them as “multiple lineage subtype cells or progenitors” without any functional experiments. This is overstretching. Fig S10 is great to reveal cells coexpressing certain markers but i) are not functional experiments to support “multi” lineage potential, ii) their frequencies do not align with FACS (FACS G3, G4 = >50%, but IF <5%?)

Their current data is sufficiently provocative without the needs to make unsupported claims. To claim that they are “progenitors”, careful functional experiments need to be performed.

2. The authors made a strong rationale to compare bulk tumor RNA seq to single cells—in fact what are the bulk tumor molecular subtype for their corresponding mouse and human tumors? Do they reflect the predominant cell population from single cell seq?

3. It is interesting that CD49f was highly expressed in epithelial cluster 5, a previously reported tumor-initiating cell marker in human MIBCs [PNAS. 2012;109:2078-83]. However, the claim that CD49f allows for “discrimination of lineage-specific population” and “lineage plasticity initiated from multiple lineage” is not backed by data.

Did they use other cell lineage additional to CD45? For instance, CD31 for endothelial cells? Based on their single cell data, CD49f is also expressed in endothelial cells (Fig 3b), it is likely their Q4 are mixed with endothelial cells that also express Sma and Axl, and Q3 may express certain CAFs (Vim and SMA are also activate myofibroblast markers, not EMT markers), since CAG-promoter is also expressed in other cell types and can carry over as GFP+ cells to further passages. Their own data in S7C showed G3 and G4 as morphologic endothelial and fibroblastic cells. I agree the existence of CK5/8+ cells (S11) but I hope the authors agree there exists technical concerns to make their provocative claims will be very confusing not carefully sorted out.

4. Fig.5 is an outstanding effort and appreciated. But we would like to caution their conclusion. For instance, it is almost impossible to sort for G1 based on their sort gate, no post-sort purity was shown which could be mistaken for their “lineage plasticity”—sort purity is a struggle for the reviewer’s own experiments (to make these claims). Same concern applies for Fig. 6C and Fig.7. It is possible plasticity occurs but not backed by current experiments [For instance, assuming their G3 and G4 are really mesenchymal (without CAFs or endothelial cells), can they show G3 mesenchymal morphology gives rise to both G2 epithelial and G3 mesenchymal morphologies?

Minor comments:

1. Their question 1 in intro: “Does the molecular subtype of an individual tumor reflect the cell of origin?” Molecular data cannot address this at all. Unless one performs RNA seq from purified basal or differentiated cells and compare those against single cancer cell seq—that still only implies their molecular similarity.

2. A lot of the supplementary figures were not referred to in text.

Reviewer #2 (Remarks to the Author):

1. The basal and luminal molecular subtypes that have been identified in bulk muscle-invasive tumors display what appear to be mutually exclusive patterns of gene expression. However, immunohistochemical analyses performed by the group at Lund have demonstrated that “urobasal”

luminal tumors consist of stratified layers of cells that maintain the protein expression patterns observed in the normal urothelium, where cells adjacent to the basement membrane express higher levels of basal proteins, cells in the middle are double-positive for basal and luminal proteins, and cells at the luminal surface express proteins associated with terminal differentiation. Therefore, the idea that bladder cancers are often heterogeneous at the single cell level is not new, and this prior work should be recognized.

2. Likewise, the idea that a population of CD49f+ cells with more “basal” phenotype that exist in heterogeneous populations of cell lines might function as bladder cancer “stem cells” that are highly tumorigenic and produce tumors that contain heterogeneous populations of cells with more basal or luminal phenotypes was first reported by Keith Chan and Irv Weissman. Even more generally, many investigators have used FACS to isolate a similar, minor cell population from conventional human breast or prostate cancer cell lines and have shown that they revert to being a minor subpopulation when they are used to form xenografts (or even when they are replated in vitro). Again, this prior work should be cited and discussed.

3. One criticism that was not adequately addressed was that the authors never did their own bulk tumor RNAseq analyses and subtype assignments to place their single cell data into context. While this is probably not absolutely necessary for the BBN tumors and cells (because other investigators have demonstrated that they are all basal), it still seems important to try to explain how the heterogeneity observed at the single cell level gives rise to a tumor with a predominant basal versus luminal gene expression bias. Again, the Lund group demonstrated using immunohistochemistry that basal tumors tend to be homogenous for basal protein expression, and this work should be discussed.

4. Specifically, which (and how many) genes were selected to represent the luminal, basal, and mesenchymal phenotypes? Is anything known about their patterns of expression in normal urothelial differentiation (or urothelial cell stratification)?

5. Recent work has demonstrated that KRT14 is a better marker for basal cells than KRT5 is because KRT5 is also expressed in intermediate cells. Would the conclusions change if KRT14 were substituted for KRT5?

Reviewer #3 (Remarks to the Author):

In this revised manuscript, the authors used scRNA-seq to examine heterogeneity in bladder cancer. Their analysis revealed that tumors contain cells with multi-lineage gene expression (basal, luminal, EMT-claudin) and suggest that these cells are in transition states between single positive lineage states. Although this is clearly an important conclusion I am not convinced that it is adequately supported. My main concern is that this could be an artefact of scRNA-seq and is the result of a cell

doublet, where more than one cell is present in the same droplet. The authors need to support these conclusions by estimating the frequency of doublets in their experiments that arise from incomplete dissociation to show that it is less of what they observe. It will also be important to confirm co-expression of all 3 basal+luminal+EMT-claudin markers in the same cell by IF. I would encourage the authors to explicitly acknowledge this potential limitation of scRNA-seq in their manuscript. Specific points are listed below:

Major points:

1) The major issue is the use of scRNA-seq to show co-expression of multi-lineage markers since there is always a chance of capturing doublets (2 or more cells are once) instead of a single cell. This happens even more often when the tissue is not completely dissociated and therefore cannot be ruled out by running a standard control such as mixed mouse/human cells. This has to be adequately addressed.

The authors claim to find cell populations expressing both basal and luminal markers based on the scRNA-seq data. They nicely support this by immunostaining for CK5+CK8 and pointing at the colocalization (fig.6a and fig.S11C). However, the authors do not provide anything to support the scRNA-seq claim that the tumors can express all 3 basal+luminal+EMT-claudin markers at once. The closest they present to supporting this claim is the colocalization with Ki67 which is misleading since a lot of cells are proliferating.

The authors have to do a triple staining with CK5+CK8+Sma(or Vim or Cldn1 etc.) to confirm the results in the tSNE.

2) The density plot is difficult to interpret with a lot of colors overlapping. Also, correlating gene expression level with co-expression of multi-lineage marker is difficult to prove in scRNA-seq. I suggest that the authors revise these claims as varying expression level is very likely a technical error associated the specific technique used (10X in this case). More than often, different scRNA-seq platforms show different expression level of a certain markers in a specific cell type even if they originate from the same tissue. Unless the authors perform the same experiment using different scRNA-seq platforms (for e.g. InDrop and Drop-seq) and show the same expression level correlating with multi-lineage marker combination, the density plots should be removed from Fig.1C and 2B.

I suggest the authors perform "GenePlot" in Seurat (or "FeatureScatter" in Seurat 3.0) instead to show the cells co-expressing Krt5 and Krt8. The authors can then select the Krt5/8 co-expressing cells and show if they express EMT-claudin markers with another GenePlot of Krt5 (or Krt8) and Vim. It would help to also have alongside I) a positive control GenePlot showing Krt8 and Krt18 co-expression (or Krt5 and Krt6a). II) a negative control GenePlot showing Krt5 and Actg2 for instance.

Other points:

- 1) Please, indicate in the results section the number of single cells sequenced in each sample from Figure 1 and 2.
- 2) The text keeps referring to the cluster number in the text. It would be easier for readers to have the tSNE with clusters alongside the tSNE in Fig.1A (or replace Fig. 1a with the tSNE labelling the cluster numbers). The Font size is also very difficult to read in each tSNE.
- 3) It seems that the macrophages are predominantly from cluster 3 not 12 as claimed in the text. Dendritic cells appear to be from cluster 3, 8, and 12 (not just 8). Monocytes seems to be in cluster 3 and 12 (not just 3). Please make the corrections in the text.
- 4) I recommend the authors to share and create a table that shows the top marker list in each population as generated from the Seurat analysis.
- 5) Based on Fig. S2, it appears that Cluster 18 does not express basal markers in the heatmap and doesn't match the claim in the text. Please comment. The number "18" in the x axis of Fig. S2 is half written.
- 6) Fig. S1C does not support the claim "We also determined that the sum of cells with high in gene expression from a single subtype exceeded the total numbers of cells, further supporting overlapping subtype expression in individual cells". The figure shows the tSNE of 357-359 bladder instead.
- 7) "Interestingly, the concomitant high expression of these genes from these three subtypes was most pronounced in the epithelial cluster 5 – a cluster high in expression for known stemness markers such as CD49f". Please show a figure to support the CD49f claim (dot plot or heatmap).
- 8) I cannot find Fig. S1D.
- 9) Add "Fig. 2C" after "In both cases, there was high, positive lineage scores (average gene expression, log nUMI) for luminal (blue), basal (orange) and emt-claudin (light green) gene expression" to support this claim.
- 10) "case 357 has high luminal subtyping and 359 has less luminal gene expression". It seems to me that 359 has less cells sequenced not less luminal gene expression. If the 359 does have less gene expression, please generate a plot which proves that epithelial cells from 357 indeed have more luminal expression than epithelial cells in 359.
- 11) The reference to fig 3D says "Q"1-2-3-4 however the fig 3D graph shows "G"1-2-3-4. Is this a mistake? If not, change the text
- 12) "Sma, Vim and Axl was observed in EpcamlowCD49low/high expressing cells (Q3)". However, Fig 3D shows that G4 (I'm assuming Q4) has more Sma and Vim than Q3 (or G3). Please explain.
- 13) Regarding the tSNEs showing expression of basal, luminal, emt-claudin, p53-like, neuroendocrine, neuronal differentiation, basal+luminal etc. Do the tSNEs show expression of a specific gene or a group of basal genes combined? The text and legend do not specify this clearly. If the tSNEs show expression of an individual basal, luminal, etc. marker please specify in the legend which specific gene is used to generate the graph (or group of genes)

14) Add all the scale bars next to the tSNE showing basal, luminal, emt, etc., expression (Fig 1B, D, Fig 2C, D). If the scale is different in each tSNE then display each individually. Add a scale bar for Fig. S2 as well. Also, the colors in the tSNE showing basal+luminal, luminal+emt-claudin, basal+emt-claudin are very faint and difficult to distinguish. Please, increase the dot size to make the colors visibly clear.

Reviewers' comments:

Reviewer #1 (Remarks to the Author):

The authors have put in considerable efforts to provide a significantly improved manuscript, with the incorporation of single cell sequencing data from n=2 BBN-induced mouse tumors and n=2 human MIBCs, much improved FACS compensation and the attempts of new in vivo tumorigenic experiments from FACS-purified populations. These are highly encouraging; however, their use of certain technical terms and concepts within the manuscripts are extremely confusing to the audience and not necessarily supported by their elegantly generated data. In brief, the terms such as lineage tracing, multi-lineage progenitors, cell-of-origin etc. are loosely used basing upon molecular data.

We appreciate the suggestions provided by this reviewer and understand that use of technical terms may be confusing for readers. As such, we have made efforts to remove and limit the use of the term's lineage tracing, multi lineage progenitor and cell of origin throughout the manuscript. Such terms have been limited to the discussion for purposes of conjecture or future directions.

1. Intra-tumor heterogeneity was reported in bladder cancer, with functionally distinct tumor subpopulations with molecular features similar to that during normal cellular differentiation [PNAS 2009;106:14016-21]. It is therefore widely accepted that a "molecular subtype" reflects the gene signature of a predominant cell type within a bulk population. These authors take an important step forward by performing single cell sequencing to identify four unique populations of CD45 neg epithelial cancer cells.

We have now incorporated and discussed the findings by Chan et al., PNAS (2009) and Volkmer et al., PNAS (2012) concerning molecular subtypes and surface antigens applied for isolating tumor initiating cell markers in human bladder cancer.

1a) Molecular subtype or expression of a single cell does not define a "lineage", we suggest to remove languages such as bi- or tri-lineage unless they can back up their "lineage tracing" claims with bar-coding type of functional lineage tracing experiments. None of the experiments in Fig 5 are considered "lineage tracing". 1b) One cannot designate a cell as co expressing "markers" based on relative mRNA expression, and directly start referring to them as "multiple lineage subtype cells or progenitors" without any functional experiments. This is overstressing. Fig S10 is great to reveal cells co expressing certain markers but i) are not functional experiments to support "multi" lineage potential, ii) their frequencies do not align with FACS (FACS G3, G4 =>50%, but IF <5%?)

The terms "bi- and tri- lineage potential" are terms that are generally reserved for genetic strategies that demark a defined cell type (or population) and subsequently allow lineage tracing through development or disease progression. Since bulk cell transplants cannot accurately define the lineage status of any individual cell (within the population) we agree that our studies do not constitute "lineage tracing". In general, identifying cells that co-express lineage markers does not mean that have multi lineage potential. Conversely, our use of the CK5-tdt genetic model is a well-defined lineage tracing approach (Fig. 7). We have removed the language of multi lineage potential from our results and reserved for conjecture during discussion.

True – clusters of cells having elevated gene expression belonging to different mRNA subtypes does not, necessarily, equate to the presence of cells co-expressing multiple lineage markers at the protein (tissue) cells. The discrepancy between gene expression analysis -> surface antigen expression and IHC detection is a problem for the field that needs to be reconciled by techniques such as MalDI toF mass spectrometry. As requested by Reviewer 3, we have now included Gene plots (Fig. 1) and triple immunofluorescence studies (Fig. 2, Fig. S8, Fig. S10) to support the presence of multi lineage expressing cells.

Their current data is sufficiently provocative without the needs to make unsupported claims. To claim that they are "progenitors", careful functional experiments need to be performed.

The reviewer is correct and we have removed the terms bi- and tri- lineage potential. These are complex lineage tracing studies requiring elegant functional modeling experimentation that may be considered for future studies. Because we have not performed genetic loss of function studies, we cannot refer to the studied populations as "progenitor". Instead, we have referred to language such as "tumor initiating cells" instead which is appropriately supported by our studies.

2. The authors made a strong rationale to compare bulk tumor RNA seq to single cells—in fact what are the bulk tumor molecular subtype for their corresponding mouse and human tumors? Do they reflect the predominant cell population from single cell seq?

Yes – both in mouse and human tumors examined, single cell analysis is supported by our bulk transcriptomic and IHC analysis. However, unlike bulk RNA, single cell analysis more clearly demonstrates the hierarchy and layering of secondary cell populations. In mouse tumors, the predominant signature is basal but with strong secondary subtypes of luminal and emt-claudin signatures. In our “bulk” transcriptomic analysis of total, mixed CD45-neg + CD45-pos mouse cells, there is a predominant basal signature with several secondary signatures including emt+claudin, squamous and luminal genes (**Fig. 3A, Fig. S11**).

With our triple immunofluorescence studies, the frequency of cells with single, double and triple lineage co-expression was dependent upon pathology grade. Specifically, regions of low grade cancer or in situ carcinoma showed more single + double marker positive cells while high grade, poorly differentiated regions showed cells with frequent triple lineage marker expression (**Fig. 2, Fig. S10**). Thus, tissue level immunotyping is essential to accompany transcriptomic analysis for proper interpretation.

Unfortunately, bulk analysis was not completed for human samples 357 and 359 due to insufficient cell numbers. However, to answer the reviewers question we have conducted IHC analysis using basal and luminal markers. We observed wide spread and high expression of CK8+ cells and a secondary population of basal (CK5, CK14) cells. These protein data matched the single cell analysis of the human samples analyzed (**Fig. 2, Fig. S10**).

3. It is interesting that CD49f was highly expressed in epithelial cluster 5, a previously reported tumor-initiating cell marker in human MIBCs [PNAS. 2012;109:2078-83]. However, the claim that CD49f allows for “discrimination of lineage-specific population” and “lineage plasticity initiated from multiple lineage” is not backed by data.

Did they use other cell lineage additional to CD45? For instance, CD31 for endothelial cells? Based on their single cell data, CD49f is also expressed in endothelial cells (Fig 3b), it is likely their Q4 are mixed with endothelial cells that also express Sma and Ax1, and Q3 may express certain CAFs (Vim and SMA are also activate myofibroblast markers, not EMT markers), since CAG-promoter is also expressed in other cell types and can carry over as GFP+ cells to further passages. Their own data in S7C showed G3 and G4 as morphologic endothelial and fibroblastic cells. I agree the existence of CK5/8+ cells (S11) but I hope the authors agree there exists technical concerns to make their provocative claims will be very confusing not carefully sorted out.

Our studies use CD49f not as a single marker but in conjunction with Epcam (to help distinguish epithelial and mesenchymal cells) and GFP (to separate host mesenchymal cells from tumorigenic EMT-like cells). To address the concern of “carry over” contributions of non-tumorigenic GFP+ cells, including CD31, from passed primary tumors, we evaluated expression levels of CD31 in tumors as a function of Epcam and transgenic GFP gating (**see data for Reviewer #1**).

Using normal mouse prostate (to obtain sufficient epithelia), we first assessed CD31 expression in Epcam-CD49f populations. We observed that total CD45-neg and CD45-neg;CD49f+ cells had detectable CD31 expression while CD49f-low cells did not. This confirms that the Reviewer is correct and that Epcam-CD49f expressing cells enrich for CD31 expression (**data for Reviewer 1, Fig. 1A**). In primary OHBBN primary tumors, we observed expression of CD31 (2-3%) in total CD45-neg cells (**Fig. 1B**). In GFP+;CD49f-high;Ep-low populations enriched for CD31 (28.4%) which was reduced >3 fold when gating on Epcam-high (8.4%) while almost no CD31+ cells (0.11%) were found in CD49f-low;Ep-low cells. As expected these data affirm that that while Epcam reduces CD31+ cells on CD49f+ gates CD31 is still abundantly expressed in OHBBN primary tumors. In OHBBN tumor transplants (MDAs propagated in GFP-neg hosts), we observed that gating on the GFP transgene (**Fig. 1C**) markedly reduced the expression of CD31 both in CD45-neg, CD45-neg;CD49f-high and CD45-neg;CD49f-high;Epcam-high populations. Collectively, these data indicate that CD31 (endothelial cells) are likely present in CD45-neg;CD49f+;Epcam-neg populations of primary tumors, however when incorporating the GFP transgenic label, there is little CD31 expression in transplants.

Evaluating the carryover of other host lineages such as CAFs (Vim+, Sma+ non-tumor cells) is more difficult to conclude primarily because of the lack of markers that distinguish normal stromal cells from EMT-like cells. While GFP+CAF5 are likely present in OHBBN induced primary tumors, we argue that, like CD31+ endothelia, they are present in minimal quantities in P0 and subsequent transplants which were used as the source of most cell isolations. Particularly, if one considers that

tumor implants typically consisted of about 10-20,000 cells growing to about $1-2 \times 10^6$ at maturity (an increase of at least 100x) at the time of implantation, total Epcam-neg cell populations (Q3 + Q4 including both normal stromal/CAFS and EMT cells) are a maximum of 5-10% of GFP+CD45-neg cells. Since the proliferative rate of normal stromal cells, including CAFs, is very low compared to that of tumor cells, we argue that the time of tumor maturation, non-transformed GFP+ stromal cells occupy less than 0.05-0.01% of the total GFP+ tumor populations (i.e. 1/100 of 5-10%).

Thus, while we cannot rule out the contribution of minor carry over contamination in our q-PCR analysis of bladder transplants, we argue that the combined use of the GFP+ markers in conjunction with the “diluting” effect achieved during the transplant of non-transformed (low or non-dividing) yields minimal (and reasonable) contamination.

4. Fig.5 is an outstanding effort and appreciated. But we would like to caution their conclusion. For instance, it is almost impossible to sort for G1 based on their sort gate, no post-sort purity was shown which could be mistaken for their “lineage plasticity”—sort purity is a struggle for the reviewer’s own experiments (to make these claims). Same concern applies for Fig. 6C and Fig.7. It is possible plasticity occurs but not backed by current experiments [For instance, assuming their G3 and G4 are really mesenchymal (without CAFs or endothelial cells), can they show G3 mesenchymal morphology gives rise to both G2 epithelial and G3 mesenchymal morphologies?

The reviewer’s point of population specificity and lineage plasticity is very well appreciated particularly with respect to purity checks which are essential for studies related to plasticity. While establishing our flow panel we conducted multiple purity checks to ensure our systems were working appropriately. From here, we have used the same sorting parameters and machine for isolation of tumor cell populations. We have included 3 representative examples from our studies (**data for Reviewer #2**). As can be seen in all gates except one gate, the purity of sorted populations was >98% (excellent by most standards!). Since few if any purity checks are 100% complete these data suggest that marked plasticity does occur particularly when considering the kinetics of “before and after analysis”

Thus, we argue that populations from G3 + G4 are efficient initiators of tumorigenesis and can revert to high expression for Epcam + CD49f. While we cannot rule out some contamination from host CAFs, tumor populations from G3 + G4 are potent initiators of tumorigenesis both in our mouse and human tumors. Such transplants efficiently give rise to G1 + G2 expressing cell populations when re analyzed from in vitro and in vivo platforms. Further, G3 + and G4 efficiently give rise to tumor outgrowth in vivo. If contamination were a primary issue with our flow cytometry technique, then these observations would be unlikely to occur in such an efficient and reproducible manner (over many trials). Moreover, we have also conducted limiting dilution studies using these populations of cells over repeated trials. The gates are designed to provide sufficient separation while also allowing for sufficient cell isolation to conduct transplants.

From these data it can be seen that

Minor comments:

1. Their question 1 in intro: “Does the molecular subtype of an individual tumor reflect the cell of origin?” Molecular data cannot address this at all. Unless one performs RNA seq from purified basal or differentiated cells and compare those against single cancer cell seq—that still only implies their molecular similarity.

We have removed this verbiage from the introduction.

2. A lot of the supplementary figures were not referred to in text.

We have referred to all supplemental information.

Reviewer #2 (Remarks to the Author):

1. The basal and luminal molecular subtypes that have been identified in bulk muscle-invasive tumors display what appear to be mutually exclusive patterns of gene expression. However, immunohistochemical analyses performed by the group at Lund have demonstrated that “urobasal” luminal tumors consist of stratified layers of cells that maintain the protein expression patterns observed in the normal urothelium, where cells adjacent to the basement membrane express higher levels of basal proteins, cells in the middle are double-positive for basal and luminal proteins, and cells at the luminal surface express proteins associated with terminal differentiation. Therefore, the idea that bladder cancers are often heterogeneous at the single cell level is not new, and this prior work should be recognized.

Yes – We have now cited studies from the Lund group that describe such comparative heterogeneity at the mRNA and protein (histological) level in bladder urothelia (PMIDs: 28195647, 29487377, 26051783). Importantly, we have emphasized findings by the Lund group suggesting supporting the “systematic disagreement in subtype classification between global mRNA and IHC profiling at the tumor cell levels” (PMID: 28195647). We concur that bulk methods cannot distinguish tumor cells and normal cells and that strategies, such as maldi tof mass spectrometry and other advanced flow cytometry techniques should be used in conjunction with gene expression analysis (single cell, global). We also appreciate that the Lund group has identified mesenchymal-like (and small cell) subtypes (PMIDs: 29487377, 26051783) which further relates to our studies concerning EMT-claudin tumor cell populations.

2. Likewise, the idea that a population of CD49f+ cells with more “basal” phenotype that exist in heterogeneous populations of cell lines might function as bladder cancer “stem cells” that are highly tumorigenic and produce tumors that contain heterogeneous populations of cells with more basal or luminal phenotypes was first reported by Keith Chan and Irv Weissman. Even more generally, many investigators have used FACS to isolate a similar, minor cell population from conventional human breast or prostate cancer cell lines and have shown that they revert to being a minor subpopulation when they are used to form xenografts (or even when they are replated in vitro). Again, this prior work should be cited and discussed

Some good suggestions - we have now cited and discussed the indicated studies by Irving Weissman's group (PMID: 19666525) and Keith Chans (PMID: 22308455) that use CD49f in conjunction with other surface antigens such as CD44, CD90 and CK20 to facilitate discrimination of poorly differentiated (basal) from differentiated human (PDX) bladder cancer cells (PMID: 22308455). We have also cited and discussed a paper showing that breast cancer stem cells transition between epithelial and mesenchymal-like (MET) cell states representing an additional example of EMT related cellular plasticity (24511467). Finally, we have included referencing to our previous studies (PMIDS: 19887604, 22350410) and from Owen Witte's group (PMIDs: 20133806, 20671189) concerning CD49f in normal prostate and prostate cancer.

3. One criticism that was not adequately addressed was that the authors never did their own bulk tumor RNA-seq analyses and subtype assignments to place their single cell data into context. While this is probably not absolutely necessary for the BBN tumors and cells (because other investigators have demonstrated that they are all basal), it still seems important to try to explain how the heterogeneity observed at the single cell level gives rise to a tumor with a predominant basal versus luminal gene expression bias. Again, the Lund group demonstrated using immunohistochemistry that basal tumors tend to be homogenous for basal protein expression, and this work should be discussed.

Correct, the Lund group has indicated that tumors with a mainly basal gene signature are homogenous for basal protein expression (PMID: 28195647) – work which we have now cited.

In our analysis of bulk RNA sequencing data, from a published data set, we identified the presence of several lineage subtypes including basal, luminal, emt-claudin and mesenchymal lineages in OHBBN induced tumors (**Fig. 3, Fig S11**). At the single cell gene expression level, we have also observed multiple lineages (**Fig. 1, Fig. S1-7**). At the protein (immunofluorescence) level, we have shown clearly and reproducibly that OHBBN tumors (FVB background) are mostly positive for basal (CK5+, CK14+, p63+) cells but have a significant population of luminal (CK8+, CK18+) and emt-claudin (CLDN3+, CLDN4+, CLDN7+) tumor cells (**Fig. 2**). While there is currently a disconnect between transcriptomic and pathology analysis of lineage marker phenotyping, the application of CyTOF mass analysis may be one approach suited to reconcile the disparity.

4. Specifically, which (and how many) genes were selected to represent the luminal, basal, and mesenchymal phenotypes? Is anything known about their patterns of expression in normal urothelial differentiation (or urothelial cell stratification)?

A challenging question. The genes used were included as **Table I** and have been drawn from other recent publications using clinical bladder cancer samples (Choi, Cancer Cell 2014; Robertson, Cell 2017). We used at least 18 luminal genes, 10 basal and 18 EMT and markers from other lineages.

As far as understanding the expression patterns of (lineage related) genes in the normal urothelial, surprisingly few investigations have carried out such analysis. This is likely a consequence of the considerable lack of reliable mouse IHC reagents needed to validate gene expression studies and the difficulties associated with obtaining sufficient quantities of normal urothelium. It may also be related to the disparity between the expression of mRNA subtyping genes and protein studies. Finally, gene sets used for clinical mRNA subtyping are, in general, not tissue specific but lineage specific. As such, such gene sets may be important for marking lineage fates in multiple organs types but not be of functional importance to urothelia. However, to address the comment we have completed an in depth PubMed literature (as best we could) search for gene function and expression in bladder (**Table for Reviewer #2**).

Of interest, however, are studies by the laboratory of Mathias Uhlen who conducted genome wide analysis to identify protein coding genes upregulated in the urinary bladder (the proteome) as compared to all other major tissue types (PMID: 26694548). Of 8,874 genes protein encoding genes identified, 0.4% were found to have elevated expression in the urinary bladder. Even with this elegant integrated -omics approach functionality could not be functionally assigned to differentiation or stratification due to technical limitations.

5. Recent work has demonstrated that KRT14 is a better marker for basal cells than KRT5 is because KRT5 is also expressed in intermediate cells. Would the conclusions change if KRT14 were substituted for KRT5?

This is interesting. To compare expression of CK5 and CK14 we did side by side staining for CK5 and CK14 in conjunction with CK8 in OHBBN induced bladder tumors. We evaluated co-expression of CK5+CK8 versus CK14+CK8 using consecutive tissue sections (low, high magnifications) and observed nearly identical staining patterns both for single positive marker expression (arrows) and double positive marker expression (arrow heads). We also obtained reagents needed to evaluate co expression of CK14+CK5 and observed predominantly overlapping expression between the two markers (**Fig. 1A, B for Reviewer #2**). At the transcriptional level we showed Epcam to demark epithelial clusters and then observed that both CK5 and CK14 were most highly expressed in epithelial clusters 3 and 8. Interestingly, we identified that CK14 is also expressed in cluster 11 where CK5 is not. Future studies may be interested in explore cluster 11 function of CK14 (**Fig. 1C**).

Reviewer #3 (Remarks to the Author):

Reviewer #3 (Remarks to the Author):

In this revised manuscript, the authors used scRNA-seq to examine heterogeneity in bladder cancer. Their analysis revealed that tumors contain cells with multi-lineage gene expression (basal, luminal, EMT-claudin) and suggest that these cells are in transition states between single positive lineage states. Although this is clearly an important conclusion I am not convinced that it is adequately supported. My main concern is that this could be an artefact of scRNA-seq and is the result of a cell doublet, where more than one cell is present in the same droplet. The authors need to support these conclusions by estimating the frequency of doublets in their experiments that arise from incomplete dissociation to show that it is less of what they observe. It will also be important to confirm co-expression of all 3 basal+luminal+EMT-claudin markers in the same cell by IF. I would encourage the authors to explicitly acknowledge this potential limitation of scRNA-seq in their manuscript.

We appreciate this reviewers detailed queries and have made our best efforts to satisfy them below as written below.

Specific points are listed below:

Major points:

1) The major issue is the use of scRNA-seq to show co-expression of multi-lineage markers since there is always a chance of capturing doublets (2 or more cells are once) instead of a single cell. This happens even more often when the tissue is not completely dissociated and therefore cannot be ruled out by running a standard control such as mixed mouse/human cells. This has to be adequately addressed.

Good suggestion - We are well aware of the potential for doublet contamination in protocols for scRNA-seq sequencing. We have taken two steps to show that our data is not an artifact of doublets, (1) During flow cytometry sorting of cells used for RNA-seq, we ensured our panel excludes doublet exclusion using two redundant gates (**Fig. S1, A**). (2) To formally quantitate detection of doublets we used the Doublet Finder Tool to show that nearly 96% of cells were singlets with 2.4% being recognized as doublet-high confidence and 1.9% as doublet-low confidence. Moreover, any possible doublets resided predominantly outside of the epithelial clusters of interest and mostly in the CD45-positive cellular fraction (**Fig. S1B**). We have also added this information to the results section.

The authors claim to find cell populations expressing both basal and luminal markers based on the scRNA-seq data. They nicely support this by immunostaining for CK5+CK8 and pointing at the colocalization (fig.6a and fig.S11C). However, the authors do not provide anything to support the scRNA-seq claim that the tumors can express all 3 basal+luminal+EMT-claudin markers at once. The closest they present to supporting this claim is the colocalization with Ki67 which is misleading since a lot of cells are proliferating. The authors have to do a triple staining with CK5+CK8+Sma(or Vim or Cldn1 etc.) to confirm the results in the tSNE.

Agree and good suggestion - Our single cell transcript data show the presence of cells that are high in expression for 1, 2 and 3 different lineage mRNA subtypes. However, to document the presence of such cells at the protein level we have followed the reviewers suggestion, obtained the needed reagents and conducted triple immunofluorescence assessing basal (CK5), luminal (CK8) and emt-claudin (Claudins 3, 4, 7) lineage markers in primary OHBBN tumors. The data now appears as **Fig. 2** where we present low and high mag images in different regions (pathology grades) and show stains for all 3 lineages including examples of single, double and triple lineage positive cells. Additional examples are shown in **Fig. S8** including data for primary and metastatic bladder cancers (**Fig. S10**).

2) The density plot is difficult to interpret with a lot of colors overlapping. Also, correlating gene expression level with co-expression of multi-lineage marker is difficult to prove in scRNA-seq. I suggest that the authors revise these claims as varying expression level is very likely a technical error associated the specific technique used (10X in this case). More than often, different scRNA-seq platforms show different expression level of a certain markers in a specific cell type even if they originate from the same tissue. Unless the authors perform the same experiment using different scRNA-seq platforms (for e.g. InDrop and Drop-seq) and show the same expression level correlating with multi-lineage marker combination, the density plots should be removed from **Fig.1C** and **2B**.

It is true that comparing expression on a biological level, while minimizing technical error, is a challenge when using Seurat V3. However, the density plots are meant to show the overlapping expression of many genes (**Table I**) from individual mRNA subtypes which we have made in transparent color to facilitate a visual for each subtype marker. The average expression (log nUMI) correlates with cells positive in the tsne plots for scRNA-seq. Thus, we do not believe values are a technical error associated with the 10x sequencing platform. However, as requested, the density plots have been removed from **Fig. 1 + 2** and moved to the supplemental information.

I suggest the authors perform “GenePlot” in Seurat (or “FeatureScatter” in Seurat 3.0) instead to show the cells co-expressing Krt5 and Krt8. The authors can then select the Krt5/8 co-expressing cells and show if they express EMT-claudin markers with another GenePlot of Krt5 (or Krt8) and Vim. It would help to also have alongside I) a positive control GenePlot showing Krt8 and Krt18 co-expression (or Krt5 and Krt6a). II) a negative control GenePlot showing Krt5 and Actg2 for instance.

Agreed - The reviewer is requesting that we show cells co-expressing specific basal (Krt5) + luminal (Krt8) combinations followed by evaluation for emt-claudin markers. To address this, we have constructed gene plots gated on Krt5-Krt8-high (mouse) cells and KRT14-KRT8 (human) cells and evaluated the presence of emt-claudin genes (Vim, Clnd3, Clnd4, Clnd7) which were compared to Krt5-Krt8-low cells (where Krt5=0 and Krt8= 0). The requested positive (Krt8-Kt18) and negative controls (Krt5-Actg2) were included. These data now appear in **Fig. 1, Fig. 3 and Fig. S7**.

Other points:

1) Please, indicate in the results section the number of single cells sequenced in each sample from Figure 1 and 2.

We have indicated the number for cells sequenced in **Fig. 1 + 2 results section**.

2) The text keeps referring to the cluster number in the text. It would be easier for readers to have the tSNE with clusters alongside the tSNE in Fig.1A (or replace Fig. 1a with the tSNE labelling the cluster numbers). The Font size is also very difficult to read in each tSNE.

Agreed – We have included a color coded legend to clarify the distinct clusters **Fig. 1A and Fig S3** which along with the labelled clusters should be clear for most readers.

3) It seems that the macrophages are predominantly from cluster 3 not 12 as claimed in the text. Dendritic cells appear to be from cluster 3, 8, and 12 (not just 8). Monocytes seems to be in cluster 3 and 12 (not just 3). Please make the corrections in the text.

We have qualified our labelling in the figures to confirm 2 major clusters for monocytes (clusters 15, 16) but a single major cluster for dendritic cells (cluster 4).

4) I recommend the authors to share and create a table that shows the top marker list in each population as generated from the Seurat analysis.

As requested we have generated tables included top-up and top-down marker lists for mouse (pooled) and each human tumor. This now appears as **Supplementary Tables 4, 5, 6**.

5) Based on Fig. S2, it appears that Cluster 18 does not express basal markers in the heatmap and doesn't match the claim in the text. Please comment. The number “18” in the x axis of Fig. S2 is half written.

The reviewer is correct, cluster 18 (old) does not express basal markers. This as well as the font size and labelling has now been corrected in revised tSNE plots (**Fig. S6**).

6) Fig. S1C does not support the claim “We also determined that the sum of cells with high in gene expression from a single subtype exceeded the total numbers of cells, further supporting overlapping subtype expression in individual cells”. The figure shows the tSNE of 357-359 bladder instead.

This sentence has been removed.

7) “Interestingly, the concomitant high expression of these genes from these three subtypes was most pronounced in the epithelial cluster 5 – a cluster high in expression for known stemness markers such as CD49f”. Please show a figure to support the CD49f claim (dot plot or heatmap).

We have included CD49f expression in **Fig. 4B** showing high expression in cluster 3 epithelial cells.

8) I cannot find Fig. S1D.

Correction – this should have been **Fig S2B**.

9) Add “Fig. 2C” after “In both cases, there was high, positive lineage scores (average gene expression, log nUMI) for luminal (blue), basal (orange) and emt-claudin (light green) gene expression” to support this claim.

Yes, this text was updated to include **Fig 3B and Fig S4B** following this sentence.

10) “case 357 has high luminal subtyping and 359 has less luminal gene expression”. It seems to me that 359 has less cells sequenced not less luminal gene expression. If the 359 does have less gene expression, please generate a plot which proves that epithelial cells from 357 indeed have more luminal expression than epithelial cells in 359.

Affirmed. We have used Ridgeplot from Seurat to show clearly that human tumor 359 has luminal gene expression than tumor 357 and not because of fewer cells (Fig. S4B).

11) The reference to fig 3D says “Q”1-2-3-4 however the fig 3D graph shows “G”1-2-3-4. Is this a mistake? If not, change the text
The reviewer is correct - this was a labelling mistake which has now been corrected so that both indicate Q (or gate).

12) “Sma, Vim and Axl was observed in EpcamlowCD49low/high expressing cells (Q3)”. However, Fig 3D shows that G4 (I’m assuming Q4) has more Sma and Vim than Q3 (or G3). Please explain.

Sma, Vim and Axl are all markedly lower in G1+G2 than G3+G4 (now Fig. 4D) but levels of Sma and Vim in G3>G4. We do not have an exact reason for this biology however we conjecture that it may be related to mesenchymal populations with qualities of stemness (i.e. CD49f high expression).

13) Regarding the tSNEs showing expression of basal, luminal, emt-claudin, p53-like, neuroendocrine, neuronal differentiation, basal+luminal etc. Do the tSNEs show expression of a specific gene or a group of basal genes combined? The text and legend do not specify this clearly. If the tSNEs show expression of an individual basal, luminal, etc. marker please specify in the legend which specific gene is used to generate the graph (or group of genes).

These plots represent multiple genes from each gene set drawn from Table I. As requested, we have now referenced Table 1 in the figures legends and explicated listed the genes for Fig. S5.

14) Add all the scale bars next to the tSNE showing basal, luminal, emt, etc., expression (Fig 1B, D. Fig 2C, D). If the scale is different in each tSNE then display each individually. Add a scale bar for Fig. S2 as well. Also, the colors in the tSNE showing basal+luminal, luminal+emt-claudin, basal+emt-claudin are very faint and difficult to distinguish. Please, increase the dot size to make the colors visibly clear.

We have added scale bars as requested in Fig 1 and Fig and in supplemental figures.

REVIEWERS' COMMENTS:

Reviewer #1 (Remarks to the Author):

We only requested the authors a small set of questions and gave the authors the freedom to rephrase certain terms to focus on the novelty of their data, yet, they insist to push points/terms not supported by their primary data.

The authors used single cell sequencing to demonstrate that there are potentially cancer cells co-expressing several markers representing different phenotypes. However, IF co-staining is not convincing, e.g. Fig 2C (1,2,3) mostly double+ instead of triple+ as they claim. Even if they are triple+, these cells might be going through a transition stage of differentiation.

They claimed to have limited the use of “lineage tracing, multilineage progenitors”, yet, the title claims “lineage heterogeneity” and “plasticity”. There are no functional data or clinical relevance to highlight a specific novel point not previously published elsewhere.

Instead of performing experiments to address our previous question #3, they simply want to argue their point through. The double negative population (Fig4c, Q3) clearly contains CAFs and other cell types, instead of using PDGFRa/b, CD49d or other markers to gate them out, they put in a long paragraph to explain. Instead of using classical EMT TFs e.g. Twist, Snail, Zeb, they used SMA as EMT markers that clearly expresses in CAFs or smooth muscle cells.

The sort purity gates are usual looking, implicating they are being gated on or manipulated by previous gates and clearly not pure. So how is it possible to prove their points in Figure 6?

More minor issues:

In the Intro section: There are certain sentences very difficult to understand: “(1) Does the molecular subtype of an individual tumor reflect the potential for tumor initiation?”: What does this mean?

“(3) Can the molecular subtype of a tumor change as a result of intrinsic or extrinsic pressure”: where is the data?

Reviewer #2 (Remarks to the Author):

Feedback on response to reviewer#2 concerns:

The authors have done an excellent job in addressing my original concerns. This is a very timely paper that will attract a lot of attention. I have only two remaining questions or comments related to the new data and discussion.

1. Emerging evidence in breast cancer and neuroendocrine prostate cancer suggests that certain differentiation-associated cell lineages may be more "plastic" than others; the prediction here would be that the more stem-like cellular subsets (basal and EMT) would be more "plastic" than the luminal ones. Do the data here support this conclusion?

2. The claudins are considered "epithelial" biomarkers - lumping them together with EMT is somewhat confusing. It would help if the authors made this point clearer in their discussion.

Feedback on response to reviewer#1 concerns:

In my opinion, the authors' most important conclusion is that epithelial "plasticity" can generate basal and luminal phenotypes in human and murine bladder cancers. A similar conclusion was advanced recently in breast cancer (Nat Commun 9:3815, 2018), and the results are consistent with other emerging evidence. This is an important conclusion that will generate substantial attention in our field.

The data are not perfect as outlined by Reviewer #1 and his colleagues (they use the word, "we", to describe their concerns). However, I think the conclusions are adequately supported by the data, and a lot of additional effort will be required to fully address the experimental question. Similar concerns were raised about "cancer stem cells", and the debate about their origins still continues to this day.

Hope this helps!

Reviewer #3 (Remarks to the Author):

Authors addressed my concerns satisfactory in their revision.

REVIEWERS' COMMENTS:

Reviewer #1 (Remarks to the Author):

We only requested the authors a small set of questions and gave the authors the freedom to rephrase certain terms to focus on the novelty of their data, yet, they insist to push points/terms not supported by their primary data.

The authors used single cell sequencing to demonstrate that there are potentially cancer cells co-expressing several markers representing different phenotypes. However, IF co-staining is not convincing, e.g. Fig 2C (1,2,3) mostly double+ instead of triple+ as they claim. Even if they are triple+, these cells might be going through a transition stage of differentiation.

They claimed to have limited the use of "lineage tracing, multilineage progenitors", yet, the title claims "lineage heterogeneity" and "plasticity". There are no functional data or clinical relevance to highlight a specific novel point not previously published elsewhere.

Instead of performing experiments to address our previous question #3, they simply want to argue their point through. The double negative population (Fig4c, Q3) clearly contains CAFs and other cell types, instead of using PDGFRa/b, CD49d or other markers to gate them out, they put in a long paragraph to explain. Instead of using classical EMT TFs e.g. Twist, Snail, Zeb, they used SMA as EMT markers that clearly expresses in CAFs or smooth muscle cells.

The sort purity gates are usual looking, implicating they are being gated on or manipulated by previous gates and clearly not pure. So how is it possible to prove their points in Figure 6?

More minor issues:

In the Intro section: There are certain sentences very difficult to understand: "(1) Does the molecular subtype of an individual tumor reflect the potential for tumor initiation?": What does this mean?

"(3) Can the molecular subtype of a tumor change as a result of intrinsic or extrinsic pressure": where is the data?

Reviewer #1

1. Without being specific, the reviewer suggests that we did not alter our phrasing. On the contrary we previously removed considerable verbiage containing "lineage tracing", "multi lineage progenitor" and "cell of origin". We have made further efforts to soften our conclusions throughout the results and discussion sections. No, our co-IF is very clear and we previously indicated that we detected cells with single, double and an abundance of triple marker positive cells (**page 5**). Also, cells are expected to be in transition which is the whole point of biological plasticity. Cell would not be expected to constitutively express high levels of multiple lineage markers.
2. The main point of our transgenic model is that donor cancer cells are positively labelled with GFP. Because of this, infiltrating and host derived CAFs (normal mesenchymal cells, fibroblasts) are removed from the plasticity analysis. We carefully conducted *in vitro* organoid assays (**Fig. 6**) (which exclude CAFs) showing that plasticity can occur in a cell autonomous manner. This doesn't mean that CAFs can't influence plasticity *in vivo* and in fact this will be an important area for future modeling. We have made comment to this point in our discussion (**page 11**).
3. We disagree, our purity checks accurately show the populations of our cells after collection all with purities >96% using regular gating. However, we have clarified that our cell populations used for lineage transplants are passed twice through the flow cytometry before doing functional assays (**page 8**)
4. **Minor:**
 - a. The phrase, ".....*molecular subtype*...." has been rephrase to ".....*lineage subtype*....." making these sentences clearer (**page 3**)
 - b. The phrase, "Can the molecular subtype of a tumor...." has been changed to "Can the lineage subtype of a tumor change in a cell autonomous manner?" (**page 3**)

Reviewer #2 (Remarks to the Author):

Feedback on response to reviewer#2 concerns:

The authors have done an excellent job in addressing my original concerns. This is a very timely paper that will attract a lot of attention. I have only two remaining questions or comments related to the new data and discussion.

1. Emerging evidence in breast cancer and neuroendocrine prostate cancer suggests that certain differentiation-associated cell lineages may be more "plastic" than others; the prediction here would be that the more stem-like cellular subsets (basal and EMT) would be more "plastic" than the luminal ones. Do the data here support this conclusion?

2. The claudins are considered "epithelial" biomarkers - lumping them together with EMT is somewhat confusing. It would help if the authors made this point clearer in their discussion.

Feedback on response to reviewer#1 concerns:

In my opinion, the authors' most important conclusion is that epithelial "plasticity" can generate basal and luminal phenotypes in human and murine bladder cancers. A similar conclusion was advanced recently in breast cancer (Nat Commun 9:3815, 2018), and the results are consistent with other emerging evidence. This is an important conclusion that will generate substantial attention in our field.

The data are not perfect as outlined by Reviewer #1 and his colleagues (they use the word, "we", to describe their concerns). However, I think the conclusions are adequately supported by the data, and a lot of additional effort will be required to fully address the experimental question. Similar concerns were raised about "cancer stem cells", and the debate about their origins still continues to this day.

Hope this helps!

Reviewer #2

1. Interesting point. Yes, our data shows that basal cells and EMT cell implants are more efficient at forming tumors. By extension, basal and EMT cells implants were also more plastic and capable of forming cells with redistribution and expression of markers. We have inserted brief text in our discussion concerning this point (**page 11, 2nd paragraph**).
2. Good point – Tight junction claudins occur at variable levels in human + mouse bladder and breast cancers. The expression of Claudins (low versus medium-high) can dictate which type of intrinsic subtype they belong to including populations of EMT-Claudin-low defined at the transcriptional level in human bladder tumors (PMIDs: 26051783, 29264194). Our immunohisto level studies detected the presence of Claudin-low and mid cells coexpressing basal, luminal and EMT lineage markers. The functional significance of these populations will need to be determined in future studies. We have inserted brief discussion to clarify this point in the discussion (**page 11, 2nd paragraph**).

Reviewer #3 (Remarks to the Author):

Authors addressed my concerns satisfactory in their revision.